# A PAC-Bayesian Generalization Bound for Equivariant Networks

**Arash Behboodi**[*]
Qualcomm AI Research, Amsterdam[†]
behboodi@qti.qualcomm.com

**Gabriele Cesa**[*]
Qualcomm AI Research, Amsterdam[†]
AMLab, University of Amsterdam
gcesa@qti.qualcomm.com

**Taco Cohen**
Qualcomm AI Research, Amsterdam[†]
tacos@qti.qualcomm.com

## Abstract

Equivariant networks capture the inductive bias about the symmetry of the learning task by building those symmetries into the model. In this paper, we study how equivariance relates to generalization error utilizing PAC Bayesian analysis for equivariant networks, where the transformation laws of feature spaces are determined by group representations. By using perturbation analysis of equivariant networks in Fourier domain for each layer, we derive norm-based PAC-Bayesian generalization bounds. The bound characterizes the impact of group size, and multiplicity and degree of irreducible representations on the generalization error and thereby provide a guideline for selecting them. In general, the bound indicates that using larger group size in the model improves the generalization error substantiated by extensive numerical experiments.

## 1 Introduction

Equivariant networks are widely believed to enjoy better generalization properties than their fully-connected counterparts by including an inductive bias about the invariance of the task. A canonical example is a convolutional layer for which the translation of the input image (over $\mathbb{R}^2$) leads to translation of convolutional features. This construction can be generalized to actions of other groups including rotation and permutation. Intuitively, the inductive bias about invariance and equivariance helps to focus on features that matter for the task, and thereby help the generalization. Recently, many works provide theoretical support for this claim mainly for general equivariant and invariant models. In this work, however, we focus on a class of equivariant neural networks built using representation theoretic tools. As in Shawe-Taylor and Wood [1996], Cohen and Welling [2016b], Weiler et al. [2018a], this approach can build general equivariant networks using irreducible representations of a group. An open question is about how to design feature spaces and combine representations. Besides, the question of impact of these choices on generalization remain. On the other hand, while it is rather well understood how to build a $G$ equivariant model when $G$ is finite, the choice of the group $G$ to consider is not always obvious. Indeed, using a subgroup $H < G$ smaller than $G$ is usually sufficient to achieve significant improvements over a non-equivariant baseline. Moreover, the symmetries $G$ of the data are often continuous, which means $G$-equivariance can not be built into the model using a group convolution design as in Cohen and Welling [2016a]. In such cases, the most successful

---

[*]equal contribution

[†]Qualcomm AI Research is an initiative of Qualcomm Technologies, Inc.

36th Conference on Neural Information Processing Systems (NeurIPS 2022).

approaches approximate it with a discrete subgroup $H < G$, e.g. see Weiler and Cesa [2019] for the $G = O(2)$ case. In addition, the choice of the equivariance group $H$ can affect either the complexity or the expressiveness of the model and, therefore, its generalization capability, depending on how the architecture is adapted. For instance, if one can not freely increase the model size, using a larger group $H$ only increases the parameters sharing but reduces the expressiveness of the model. For this reason, using the whole group $G$ might not be beneficial even when feasible. In that line, we consider how the choice of group size affects generalization.

**Contributions**  In this paper, we utilize PAC Bayes framework to derive generalization bounds on equivariant networks. Our focus is on equivariant networks built following Weiler et al. [2018a], Cohen and Welling [2016b]. We combine the representation theoretic framework of Weiler and Cesa [2019] with PAC-Bayes framework of Neyshabur et al. [2018] to get generalization bounds on equivariant networks as a function of irreducible representations (irreps) and their multiplicities. Different from previous PAC-Bayes analysis, we derive the perturbation analysis in Fourier domain. Without this approach, as we will show, the naive application of Neyshabur et al. [2018] lead to degenerate bounds. As part of the proof, we derive a tail bound on the spectral norm of random matrices characterized as direct sum of irreps. Note that for building equivariant networks, we need to work with real representations and not complex ones, which are conventionally studied in representation theory. The obtained generalization bound provide new insights about the effect of design choices on the generalization error verified via numerical results. To the best of our knowledge, this is the first generalization bound for equivariant networks with an explicit equivariant structure. We conduct extensive experiments to verify our theoretical insights, as well as some of the previous observations about neural networks.

## 2   Related Works

**Equivariant networks.** In machine learning, there has been many works on incorporating symmetries and invariances in neural network design. This class of models has been extensively studied from different perspectives Shawe-Taylor [1993, 1989], Kondor and Trivedi [2018], Cohen et al. [2018], Mallat [2012], Dieleman et al. [2016], Cohen and Welling [2016a,b], Worrall et al. [2017], Weiler et al. [2018b,a], Thomas et al. [2018], Bekkers et al. [2018], Weiler and Cesa [2019], Bekkers [2020], Defferrard et al. [2019], Finzi et al. [2020], Weiler et al. [2021], Cesa et al. [2022] to mention only some. Indeed, the most recent developments in the field of equivariant networks suggest a design which enforces equivariance not only at a global level Laptev et al. [2016] but also at each layer of the model. An interesting question is how a particular choice for designing equivariant networks affect its performance and in particular generalization. A related question is whether including this inductive bias helps the generalization. The authors in Sokolić et al. [2017a] consider general invariant classifiers and obtain robustness based generalization bounds, as in Sokolić et al. [2017b], assuming that data transformation changes the inputs drastically. For finite groups, they reported a scaling of generalization error with $1/\sqrt{|H|}$ where $|H|$ is the cardinality of underlying group. In Elesedy [2022], the gain of invariant or equivariant hypotheses was studied in PAC learning framework, where the gain in generalization is attributed to shrinking the hypothesis space to the space of orbit representatives. A similar argument can be found in Sannai et al. [2021], where it is shown that the invariant and equivariant models effectively operate on a shrunk space called Quotient Feature Space (QFS), and the generalization error is proportional to its volume. They generalize the result of Sokolić et al. [2017a] and relax the robustness assumption, although their bound shows suboptimal exponent for the sample size. Similar to Sokolić et al. [2017a], they focus on scaling improvement of the generalization error with invariance and do not consider computable architecture dependent bounds. The authors in Lyle et al. [2020] use PAC-Bayesian framework to study the effect of invariance on generalization, although do not obtain any explicit bound. Bietti and Mairal [2019] studies the stability of models equivariant to compact groups from a RKHS point of view, relating this stability to a Rademacher complexity of the model and a generalization bound. The authors in Elesedy and Zaidi [2021] provided the generalization gain of equivariant models more concretely and reported VC-dimension analysis. Subsequent works also considered the connection of invariance and generalization Zhu et al. [2021]. In contrast with these models, we consider the equivariant networks with representation theoretic construction. Beyond generalization, we are interested in getting design insights from the bound.

**Generalization error for neural networks.** A more general study of generalization error in neural networks, however, has been ongoing for some time with huge body of works on the topic. We refer to some highlights here. A challenge for this problem was brought up in Zhang et al. [2017], where it was shown using the image-net dataset with random labels, that the generalization error can be arbitrarily large for neural networks, as they achieve small training error but naturally cannot generalize because of random labels of images. Therefore, any complexity measure for neural networks should be consistent with the above observation. As a result, uniform complexity measures like VC-dimension do not satisfy this requirement as they provide a uniform measure over the whole hypothesis space.

In this light, recent works related the generalization errors to quantities like margin and different norms of weights for example in, among others, Wei and Ma [2019], Sokolić et al. [2017b], Neyshabur et al. [2018], Arora et al. [2018], Bartlett et al. [2017], Golowich et al. [2018], Dziugaite and Roy [2018a], Long and Sedghi [2019], Vardi et al. [2022], Ledent et al. [2021], Valle-Pérez and Louis [2020]. Some of these bounds are still vacuous or dimension-dependent (see Jiang et al. [2020] for detailed experimental investigation and Nagarajan and Kolter [2019], Koehler et al. [2021], Negrea et al. [2021] for follow-up discussions on uniform complexity measures). Class of convolutional neural networks are particularly relevant as they can be considered as a special case of equivariant models. These models are discussed in Pitas et al. [2019], Long and Sedghi [2019], Vardi et al. [2022], Ledent et al. [2021]. We choose PAC Bayesian framework for deriving generalization bounds. There are many works on PAC Bayesian bounds for neural networks Neyshabur et al. [2018], Biggs and Guedj [2022], Dziugaite and Roy [2017, 2018b,a], Dziugaite et al. [2020] ranging from randomized and deterministic bounds to choosing priors for non-vacuous bounds. Our method combines the machinery of Neyshabur et al. [2018] with representation theoretic tools.

## 3 Background

### 3.1 Preliminaries and notation

We fix some notations first. We consider a classification task with input space $\mathcal{X}$ and output space $\mathcal{Y}$. The input is assumed to be bounded in $\ell_2$-norm by $B$. The data distribution, over $\mathcal{X} \times \mathcal{Y}$, is denoted by $\mathcal{D}$. The hypothesis space, $\mathcal{H}$, consists of all functions realized by a $L$-layer general neural network with 1-Lipschitz homogeneous activation functions[3]. The network function, $f_{\mathbb{W}}(\cdot)$ with fixed architecture, is specified by its parameters $\mathbb{W} := (\boldsymbol{W}_1, \dots, \boldsymbol{W}_L)$. The operations on $\mathbb{W}$ are extended from the underlying vector spaces of weights. For any function $f(\cdot)$, the $k$'th component of the function is denoted by $f(\boldsymbol{x})[k]$ throughout the text. For a loss function $\mathsf{L} : \mathcal{H} \times \mathcal{X} \times \mathcal{Y} \to \mathbb{R}$, the empirical loss is defined by $\hat{\mathcal{L}}(f) := \frac{1}{m} \sum_{i=1}^{m} \mathsf{L}(f, (\boldsymbol{x}_i, y_i))$ and the true loss is defined by $\mathcal{L}(f) = \mathbb{E}_{(\boldsymbol{x}, y) \sim \mathcal{D}}[\mathsf{L}(f, (\boldsymbol{x}, y))]$. For classification, the margin loss is given by

$$\mathcal{L}_\gamma(f_{\mathbb{W}}) = \mathbb{P}_{(\boldsymbol{x}, y) \sim \mathcal{D}} \left( f_{\mathbb{W}}(\boldsymbol{x})[y] \leq \gamma + \max_{j \neq y} f_{\mathbb{W}}(\boldsymbol{x})[j] \right). \tag{1}$$

The true loss of a given classifier is then given by $\mathcal{L}(f_{\mathbb{W}}) = \mathcal{L}_0(f_{\mathbb{W}})$. The empirical margin loss is similarly defined and denoted by $\hat{\mathcal{L}}_\gamma$. In this work, we consider invariance with respect to transformations modeled as the action of a compact group $G$. See Supplementary 7 for a brief introduction to group theory and the concepts that we will use throughout this paper.

### 3.2 Equivariant Neural networks

We provide a concise introduction to equivariant networks here with more details given in Supplementary 8. We adopt a representation theoretic framework for building equivariant models.

**Example.** Consider a square integrable complex valued function $f$ with bandwidth $B$ defined on 2D-rotation group $SO(2)$ and represented using its Fourier series as $f(\theta) = \sum_{n=0}^{B} a_n e^{in\theta}$. Consider an equivariant linear functional of $f$ mapping it to another function on $SO(2)$. In Fourier space, the linear transformation is given by $\boldsymbol{W}\boldsymbol{a}$ with $\boldsymbol{a}$ as the Fourier coefficient vector $\boldsymbol{a} = (a_0, \dots, a_B)$. The action of rotation group on the input and output space is simply given by $f(\theta) \to f(\theta - \theta_0)$ with $\theta_0$ as the rotation angle, or equivalently in Fourier space as a linear transformation $\boldsymbol{a} \to \rho(\theta_0)\boldsymbol{a}$

---

[3]The function $\sigma(\cdot)$ is homogeneous if and only if for all $\boldsymbol{x}$ and $\lambda \in \mathbb{R}$, we have $\sigma(\lambda \boldsymbol{x}) = \lambda \sigma(\boldsymbol{x})$.

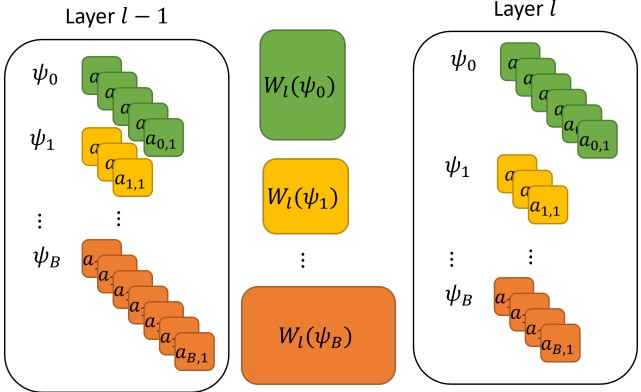

Figure 1: An equivariant layer maps the coefficients corresponding to each frequency (irrep) $\psi$ to a new set of coefficients based on the matrix $\widehat{W}_l(\psi)$, which include blocks of $\widehat{W}_l(\psi, i, j)$ with $i \in [m_{l-1,\psi}]$ and $j \in [m_{l,\psi}]$.

with $\rho(\theta_0) = \mathrm{diag}\left(1, e^{-i\theta_0} \dots e^{-iB\theta_0}\right)$. Since it holds for all $\boldsymbol{a}$, Equivariance condition means $\boldsymbol{W}\rho(\theta_0) = \rho(\theta_0)\boldsymbol{W}$ for all $\theta_0$, which implies that $\boldsymbol{W}$ should be diagonal. A general feature space can be built by stacking multiple linear transformations $\boldsymbol{W}$, and more flexibly, by letting the transformations acting only on some of the frequencies (for example, a transformation that acts only on $a_B$ for frequency $B$). This machinery can be generalized to represent equivariant networks with respect to any compact group, by using irreducible representations (irreps) of the group in place of complex exponential $e^{in\theta}$ of this example. The key is to notice that $e^{in\theta}$'s are irreps of $SO(2)$.

**General Model.** Consider an Multi-Layer Perceptron (MLP) with $L$ layers with the layer $l \in [L]$ given by the matrix $\boldsymbol{W}_l \in \mathbb{R}^{c_l \times c_{l-1}}$. The action of the group $H$ on layer $l$ is determined by a linear transformation $\rho_l$, which is called the group representation on $\mathbb{R}^{c_l}$ (see section 8 for more details). A group element $h \in H$ linearly transforms the layer $l$ by the matrix $\rho_l(h)$. The matrix $\boldsymbol{W}_l$ is equivariant w.r.t. the representations $\rho_l$ and $\rho_{l-1}$ acting on its input and output if and only if for all $h \in H$, we have:

$$\boldsymbol{W}_l\rho_{l-1}(h) = \rho_l(h)\boldsymbol{W}_l \quad .$$

Representation theory provides a way of characterizing equivariant layers using Maschke's theorem and Schur's lemma. The key step is to start from characterizing the irreducible representations (irreps) $\{\psi\}$ of the group $H$. As we will see, irreps are closely related to generalization of Fourier analysis for functions defined on $H$. Maschke's theorem implies that the representation $\rho_l$ decomposes into direct sum of irreps as $\rho_l = Q_l \left(\bigoplus_\psi \bigoplus_{i=1}^{m_{l,\psi}} \psi\right) Q_l^{-1}$, where $m_{l,\psi}$ is the number of times the irrep $\psi$ is present in the representation $\rho_l$, that is, its multiplicity. Each $\psi$ is a $\dim_\psi \times \dim_\psi$ matrix, and the direct sum is effectively a block diagonal matrix with matrix irreps $\psi$ on the diagonal. Through this decomposition, we can parameterize equivariant networks in terms of irreps, that is in Fourier space. Defining $\widehat{W}_l = Q_l^{-1}\boldsymbol{W}_l Q_{l-1}$, the equivariance condition writes as

$$\widehat{W}_l \left(\bigoplus_\psi \overset{m_{l-1,\psi}}{\underset{i=1}{\bigoplus}} \psi\right) = \left(\bigoplus_\psi \overset{m_{l,\psi}}{\underset{i=1}{\bigoplus}} \psi\right) \widehat{W}_l.$$

The block diagonal structure of $\left(\bigoplus_\psi \bigoplus_{i=1}^{m_{l,\psi}} \psi\right)$ induces a similar structure on $\widehat{W}_l$ (see Figure 6). By $\widehat{W}_l(\psi_2, j, \psi_1, i)$ denote the block in $\widehat{W}$ that relates $i$'th instance of $\psi_1$ to $j$'th instance of $\psi_2$, with $i \in [m_{l-1,\psi_1}]$ and $j \in [m_{l,\psi_2}]$, as:

$$\forall h, \quad \widehat{W}_l(\psi_2, j, \psi_1, i)\psi_1(h) = \psi_2(h)\widehat{W}_l(\psi_2, j, \psi_1, i).$$

Schur's lemma helps us to characterize the equivariant kernels. Note for neural network implementation, we need to work with a version of Schur's lemma for real-valued representations given in Supplementary 8. Schur's lemma implies that if $\psi_1 \neq \psi_2$, the block needs to be zero to guarantee equivariance. Otherwise, it needs to have one of the three forms in Supplementary 8, depending on

the *type* of $\psi_1$. To simplify the notation, we write $\widehat{\boldsymbol{W}}_l(\psi, j, \psi, i)$ as $\widehat{\boldsymbol{W}}_l(\psi, j, i)$. As it is shown in Supplementary 8, the matrix $\widehat{\boldsymbol{W}}_l(\psi, j, i)$ can be written as the linear sum

$$\widehat{\boldsymbol{W}}_l(\psi, j, i) = \sum_{k=1}^{c_\psi} \widehat{w_{l,i,j}}(\psi)_k \boldsymbol{B}_{l,\psi,i,j,k} \tag{2}$$

where $\{\widehat{w_{l,i,j}}(\psi)_k\}_{k=1}^{c_\psi}$ are learnable parameters with fixed matrices $\boldsymbol{B}_{l,\psi,i,j,k}$. The value of $c_\psi$ is either 1, 2 or 4. The structure of $\boldsymbol{B}_{l,\psi,i,j,k}$ changes according to each type $c_\psi$. Each layer is parameterized by the matrices $\widehat{\boldsymbol{W}}_l(\psi)$, which include $c_\psi$ parameters $\widehat{\boldsymbol{w}_{l,i,j}}(\psi) \in \mathbb{R}^{c_\psi}$ with $i \in [m_{l-1,\psi}]$ and $j \in [m_{l,\psi}]$. Therefore, for each layer, there will be $\sum_\psi m_{l,\psi} m_{l-1,\psi} c_\psi$ parameters with the width of the layer $l$ given as $c_l = \sum_\psi m_{l,\psi} \dim_\psi$. Note that, if $\boldsymbol{x}_l = \boldsymbol{W}_l \boldsymbol{x}_{l-1}$, then $\widehat{\boldsymbol{x}}_l = \widehat{\boldsymbol{W}}_l \widehat{\boldsymbol{x}}_{l-1}$, where $\widehat{\boldsymbol{x}}_l = Q_l^{-1} \boldsymbol{x}_l$. The non-linearities should satisfy equivariance condition to have full end-to-end equivariance. An important question is which representation $\rho_l$ of $H$ should be used for each layer. For the input, the representation is fixed by the input space and its structure. However, the intermediate layers can choose $\rho_l$ by selecting irreps and their multiplicity. We explain how this choice impacts generalization bounds. We consider a general equivariant network denoted by $f_\mathbb{W}$ with the parameters of $l$'th layer given by $\widehat{\boldsymbol{w}_{l,i,j}}(\psi)$ for all irreps $\psi$, $i \in [m_{l-1,\psi}]$ and $j \in [m_{l,\psi}]$.

## 4 PAC-Bayesian bound

We derive PAC-Bayes bounds for equivariant networks $f_\mathbb{W}$ presented in the previous section. Although the proof follows a similar strategy as in Neyshabur et al. [2018], it differs in two important aspects. First, the weights of equivariant MLPs have specific structure requiring reworking proof steps. Second, we carry out the analysis in Fourier domain. In Supplementary 11, we also show that naively applying norm-bounds of Neyshabur et al. [2018] cannot explain the generalization behaviour of equivariant networks. A simple version of our result is given below.

**Theorem 4.1** (Homogeneous Bounds for Equivariant Networks). *For any equivariant network, with high probability we have:*

$$\mathcal{L}(f_\mathbb{W}) \leq \hat{\mathcal{L}}_\gamma(f_\mathbb{W}) + \tilde{\mathcal{O}} \left( \sqrt{\frac{\prod_l \|\boldsymbol{W}_l\|_2^2}{\gamma^2 m \eta} \left(\sum_{l=1}^L \sqrt{M(l,\eta)}\right)^2 \left(\sum_l \frac{\sum_{\psi,i,j} \left\|\widehat{\boldsymbol{W}}_l(\psi,i,j)\right\|_F^2 / \dim_\psi}{\|\boldsymbol{W}_l\|_2^2}\right)} \right)$$

*where $\eta \in (0, 1)$ and*

$$M(l, \eta) := \log\left(\frac{\sum_{l=1}^L \sum_\psi m_{l,\psi}}{1 - \eta}\right) \max_\psi \left(5 m_{l-1,\psi} m_{l,\psi} c_\psi\right). \tag{3}$$

The complete version of the theorem is given in Theorem 9.1.

We comment briefly on the proof steps. PAC-Bayesian generalization bounds start with defining a prior $P$ and posterior $Q$ over the network parameters. The network is drawn randomly using $Q$, which is also used to evaluate the average loss and generalization error. The PAC-Bayesian bound provides a bound on the average generalization error of networks randomly drawn from $Q$ (Section 4.1). The next step is to de-randomize the bound and to obtain the generalization error for a specific instance $f_\mathbb{W} \in \mathcal{H}$. Among different strategies, we follow the perturbation based method of Neyshabur et al. [2018]. The main idea is to carefully define a posterior $Q$, such that the networks randomly drawn from $Q$ have small *distance* with the specific $f_\mathbb{W}$, and therefore the loss averaged over $Q$ can be used to bound the loss of $f_\mathbb{W}$. One way to choose $Q$ is to define a Gaussian distribution around the parameters $\mathbb{W}$ and choose the variance to control the output perturbation (Section 4.2).

In what follows we provide more details about these steps.

### 4.1 PAC-Bayesian Bounds for Randomized Networks

The starting point is the PAC-Bayes theorem Langford and Shawe-taylor [2002], McAllester [1998], which provides a generalization bound on randomized classifiers. If the function is chosen randomly

from a distribution $Q$, then define the average losses as $\mathcal{L}(Q) = \mathbb{E}_{f \sim Q} \mathcal{L}(f)$ and $\hat{\mathcal{L}}(Q) = \mathbb{E}_{f \sim Q} \hat{\mathcal{L}}(f)$. We use the following version from Germain et al. [2009].

**Lemma 4.2.** *For any hypothesis space $\mathcal{H}$, let $P$ be a probability distribution on $\mathcal{H}$. Then for all $\delta \in (0, 1]$, with probability $1 - \delta$, for all $Q$ on $\mathcal{H}$, we have:*

$$\mathcal{L}(Q) \leq \hat{\mathcal{L}}(Q) + \sqrt{\frac{D(Q\|P) + \log(\xi(m)/\delta)}{2m}}. \tag{4}$$

*where $\xi(m) := \sum_{k=0}^{m} \binom{m}{k} (k/m)^k (1 - k/m)^{m-k}$.*

In PAC-Bayes vernacular, the distributions $P$ and $Q$ are called *a prior* and *a posterior distribution* on $\mathcal{H}$. This indicates that the distribution $P$ and $Q$ are chosen independently with $Q$ typically chosen based on the training data and the obtained model, and $P$ chosen independently based on any information available before training. Choosing an appropriate prior plays a central role in getting non-vacuous bounds. For instance in Dziugaite and Roy [2017], the prior $P$ is chosen using a separate dataset not used during training. Lemma 4.2 provides immediately a generalization bound for randomized equivariant classifiers. The hypothesis space is parameterized by the network parameters $\mathbb{W}$, which contains kernel parameters $\widehat{\boldsymbol{w}_{l,i,j}}(\psi)$. We choose a prior distribution $P$ as zero-mean normal with covariance matrix $\sigma^2 \boldsymbol{I}$ over the parameters. The posterior $Q$ is chosen as normal distribution with the final parameters $\widehat{\boldsymbol{w}_{l,i,j}}(\psi)$ as mean value and the same covariance matrix $\sigma^2 \boldsymbol{I}$. The KL-divergence in the PAC-Bayesian theorem is then given by:

$$D(Q\|P) = \frac{\sum_{l,\psi,i,j} \left\|\widehat{\boldsymbol{w}_{l,i,j}}(\psi)\right\|_2^2}{2\sigma^2}. \tag{5}$$

The generalization bound depends on the sum of the norm of kernels. In the next step, we de-randomize the bound to get a bound on the generalization error of $f_{\mathbb{W}}$.

## 4.2 De-Randomization and Perturbation Bounds

Our de-randomization technique follows closely that of Neyshabur et al. [2018]. We provide the sketch of derivations here, and the full proof is relegated to the supplementary materials. To de-randomize the previous bound, a common step is to choose $Q$ such that the probability of margin violation is controlled. Let $\mathcal{S}_{\mathbb{W}}$ be defined as:

$$\mathcal{S}_{\mathbb{W}} := \{h \in \mathcal{H} : \|h - f_{\mathbb{W}}\|_\infty \leq \gamma/4\}. \tag{6}$$

Any function on this set can change the margin of $f_{\mathbb{W}}$ at most by $\gamma/2$. Therefore, for any $h$, $\mathcal{L}(f_{\mathbb{W}}) \leq \mathcal{L}_{\gamma/2}(h)$, which implies $\mathcal{L}(f_{\mathbb{W}}) \leq \mathcal{L}_{\gamma/2}(Q)$ if $Q$ is supported only on this set. Similarly $\hat{\mathcal{L}}_{\gamma/2}(Q) \leq \hat{\mathcal{L}}_\gamma(f_{\mathbb{W}})$. To choose $Q$, first, we characterize the output perturbation of equivariant networks for a given input perturbation. Next, we determine the variance $\sigma$ such that the output perturbation is bounded by $\gamma/4$ with probability[4] $1/2$. The first step is the perturbation analysis.

**Lemma 4.3** (Perturbation Bound). *For a neural network $f_{\mathbb{W}}(\cdot)$ with input space of $\ell_2$-norm bounded by $B$, and for any weight perturbations $\mathbb{U} = (\boldsymbol{U}_1, \ldots, \boldsymbol{U}_L)$, we have:*

$$\|f_{\mathbb{W}+\mathbb{U}}(\boldsymbol{x}) - f_{\mathbb{W}}(\boldsymbol{x})\|_2 \leq eB \left(\prod_{i=1}^{L} \|\boldsymbol{W}_i\|_2\right) \sum_{i=1}^{L} \frac{\|\boldsymbol{U}_i\|_2}{\|\boldsymbol{W}_i\|_2}. \tag{7}$$

*For equivariant networks, where the weights are parameterized as in eq. 2, the spectral norm of the perturbation $\boldsymbol{U}_l$ is bounded as*

$$\|\boldsymbol{U}_l\|_2 \leq \sqrt{\max_\psi \max_{1 \leq i \leq m_{l-1,\psi}} m_{l-1,\psi} \frac{1}{\dim_\psi} \sum_{j=1}^{m_{l,\psi}} \left\|\widehat{\boldsymbol{U}_l}(\psi, j, i)\right\|_F^2} \tag{8}$$

The first inequality is already given in Neyshabur et al. [2018]. The inequality 8 will be shown in supplementary materials.

---

[4] This value can be changed to any probability with proper adjustments. See the supplementary materials.

The perturbation bound already suggests the benefit of equivariant Kernels. As it will be shown in Supplementary materials, the equivariant kernels w.r.t. $\psi$ satisfy

$$\|\boldsymbol{W}\|_2 = \frac{1}{\sqrt{\dim_\psi}} \|\boldsymbol{W}\|_F, \tag{9}$$

which is the smallest possible norm (compare with general relation for $\dim_\psi \times \dim_\psi$ matrices: $\|\boldsymbol{W}\|_2 \geq \frac{1}{\sqrt{\dim_\psi}} \|\boldsymbol{W}\|_F$). Therefore, the term $\frac{1}{\dim_\psi} \left\| \widehat{\boldsymbol{U}_l}(\psi, j, i) \right\|_F^2$ in the perturbation bound is already tight.

The perturbation bound helps defining a posterior $Q$ by adding zero-mean Gaussian perturbations $\widehat{u_{l,i,j}}(\psi)_k$ of variance $\sigma^2$ to the learnable parameters $\widehat{w_{l,i,j}}(\psi)_k$. The output perturbation is given by the above theorem in terms of norm of $\boldsymbol{U}_i$. The following lemma controls the norm of this random matrix, and thereby the output perturbation.

**Lemma 4.4.** *Consider a random equivariant matrix $\boldsymbol{U}_l$ defined by i.i.d. Gaussian $N(0, \sigma^2)$ choice of $\widehat{u_{l,i,j}}(\psi)_k$ in eq. 2. We have:*

$$\mathbb{P}\left( \|\boldsymbol{U}_l\|_2 \geq \sigma \sqrt{\max_\psi \left(5 m_{l-1,\psi} m_{l,\psi} c_\psi t\right)} \right) \leq \left( \sum_\psi m_{l,\psi} \right) e^{-t} \tag{10}$$

The proof is given in supplementary materials. With Lemma 4.4, we can choose the variance $\sigma^2$ such that the output perturbation does not violate the intended margin. Using an union bound over all layers, one finds that by choosing $t = \log\left( 2 \sum_l \sum_\psi m_{l,\psi} \right)$, with probability at least $\frac{1}{2}$ the following inequality holds for every layer $l \in [L]$:

$$\|\boldsymbol{U}_l\|_2 < \sigma \sqrt{\max_\psi 5 m_{l-1,\psi} m_{l,\psi} c_\psi} \sqrt{\log\left( 2 \sum_l \sum_\psi m_{l,\psi} \right)} \tag{11}$$

We can use this bound jointly with the perturbation bound eq. 7 to choose $\sigma$ such that the margin is below $\gamma/4$ with high probability. Note that the $\sigma$ chosen in this way is a function of learned weights. This cannot be the case because of the prior $P$ should be fixed prior to learning. A trick for circumventing this issue is to select many priors that can adequately cover the space of possible weights, and take the union bound over it. The complete proof of the theorem is given in Supplementary 9, where we also consider special cases of this generalization bound for group convolutional networks Cohen and Welling [2016a].

**Remarks on the generalization error from the theory.** The generalization error of Theorem 4.1 contains multiple terms. First, the term $\eta$ is a hyperparameter, which will be fixed to $1/2$ for the rest. The term $M(l, \eta)$ encodes the multiplicity and type of used irreps and their impact on the generalization error. Let the hidden layer dimension $c_l = \sum_\psi \dim_\psi m_{l,\psi}$ be fixed to a constant. For Abelian groups, all real irreps are at most 2-dimensional, therefore, restricting the network to 2-dimensional irreps, we have:

$$M(l, \eta) = \log\left( \frac{\sum_{l=1}^{L} c_l}{2(1-\eta)} \right) \max_\psi \left(5 m_{l-1,\psi} m_{l,\psi} c_\psi\right).$$

The bound can be controlled further by appropriate choice of $m_{l-1,\psi}, m_{l,\psi}, c_\psi$, which favors smaller multiplicity and type $c_\psi$. Therefore, the bound favors using different irreps instead of repeating them. For non-abelian cases, irreps can have larger $\dim_\psi$, which keeping the layer dimension $c_l$ fixed, amounts to smaller multiplicity. This can improve the sum $\sum_{l=1}^{L} \sum_\psi m_{l,\psi}$ in $M(l, \eta)$ and indicates a potential improved generalization for non-abelian groups. For dihedral groups, which is in general non-abelian, the real irreps are also at most two dimensional, however, of type $c_\psi = 1$. Note that for finite groups, there is a connection between irrep degrees and group order $|H| = \sum_\psi \dim_\psi^2 / c_\psi$. The impact of group size manifests itself in the whole generalization error term. We numerically verify this in the experimental results. The inverse dependence on margin $\gamma$ is expected and similar to other

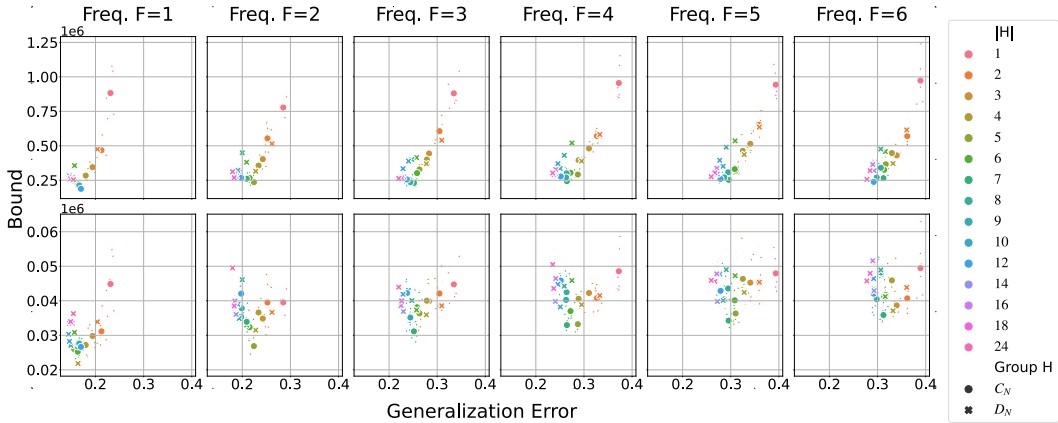

Figure 2: Comparison between the bound in Theorem 4.1 (1st row) and the one in Supplementary 11 (2nd row) on the synthetic O(2) datasets with different frequencies. The bound from Supplementary 11 clearly does not capture the effect of group equivariance.

works for instance Neyshabur et al. [2018]. The rest of the terms contain different norms of neural network kernels. We discuss them more in connection with Neyshabur et al. [2018].

**Comparison to Neyshabur et al. [2018].** There are two main differences with the bound in Neyshabur et al. [2018]. First, their bound has a norm dependence of $\tilde{\mathcal{O}}\left((\prod_l \|\boldsymbol{W}\|_2^2)(\sum_l \|\boldsymbol{W}\|_F^2 / \|\boldsymbol{W}\|_2^2)\right)$, while in our result, the Frobenius norm includes an additional scaling of $1/\dim_\psi$ in $\left\|\widehat{\boldsymbol{W}}_l(\psi, i, j)\right\|_F^2 / \dim_\psi$. This term is strictly smaller in our bound for $\dim_\psi > 1$. The second difference is the term $\left(\sum_l M(l, \eta)\right)^2$, which is replaced with $L^2 h \ln(Lh)$ where $h$ is the layer width. In our case, the layer width is $c_l = \sum_\psi \dim_\psi m_{l,\psi}$. Choosing all layer widths $c_l$ equal to $h$, we recover the $\ln(Lh)$ term from Neyshabur et al. [2018] in our bound. The remaining term is $O\left(\left(\sum_l \sqrt{m_{l-1,\psi} m_{l,\psi} c_\psi}\right)^2\right)$. As an example, assume the same multiplicity $m_{l,\psi} = m_l$ and same dimension $\dim_\psi = \dim$ for all irreps, that is, $h = N \dim m_l$ with $N$ being the number of used irreps. Then, we get a term as $O\left(h^2 L^2/(N \dim)^2\right)$. As long as the multiplicity $m_l$ is smaller than $N \dim$, i.e. number of used irreps times their dimension, our bound is strictly tighter. Intuitively, this means that using more diverse irreps leads to better generalization. See Section 11 for more detailed discussions.

## 5 Experiments

In this section, we also validate the theoretical insights derived in the previous section. More experiments and discussions are included in Supplementary 12.

We have used datasets based on natural images and synthetic data. In all experiments, we consider data symmetries $G$ which are subgroups of the group $O(2)$ of planar rotations and reflections. This includes a number of finite/continuous and commutative/non-commutative groups, in particular $C_N$ ($N$ rotations), $D_N$ ($N$ rotations and reflections), $SO(2)$ (continuous planar rotations) and $O(2)$ itself (see Supplementary 7). Moreover, all models are based on group convolution over a finite subgroup $H < G$ Cohen and Welling [2016a]. This means that the features of the layer $l$ have shape $|H| \times c_l^H$, that is, $c_l^H$ channels of dimension $|H|$. For each layer $l$, we keep the total number of features (approximatively) constant when changing the group $H$. Models are trained until 99% of the training set is correctly classified with at least a margin $\gamma$. We used $\gamma = 10$ in the synthetic datasets and $\gamma = 2$ in the image ones.

First, we compare our result with the bound derived in Supplementary 11 using naive application of Neyshabur et al. [2018]. In Fig. 2 and Fig. 3, we compare these two bounds on 6 synthetic datasets with $O(2)$ and $SO(2)$ symmetries (one per column). Each dataset lives on high dimensional tori and $O(2)$ (or $SO(2)$) rotates the circles composing them, with different frequencies. We consider

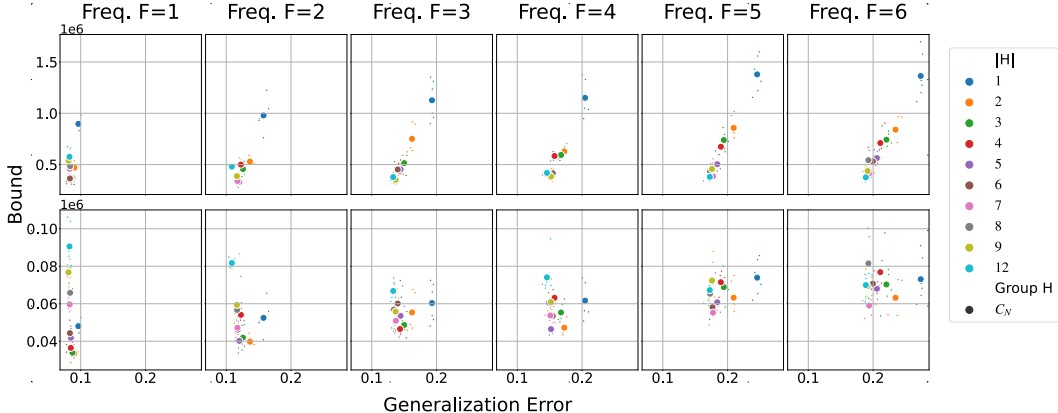

Figure 3: Comparison between Theorem 4.1 (1st row) and the one in Supplementary 11 (2nd row) on the synthetic SO(2) datasets.

different maximum frequencies $F \in \{1, 2, \ldots, 6\}$ in the experiments. For example, in Fig. 2, the first plot on the left corresponds to a synthetic dataset with O(2) symmetry generated only using frequency 1. See Supplementary 12.1 for more details and visualization of the synthetic data.

The results in Fig. 2 and 3 indicate that the bound built using the strategy from Neyshabur et al. [2018] cannot explain the effect of equivariance on generalization. Conversely, the bound in Theorem 4.1, which uses the parametrization in the Fourier domain, correlates with the measured generalization error, suggesting that it can account for the effect of different groups on generalization. In addition, note that it also correlates with the saturation effect observed when using larger discrete groups $H$ to approximate the continuous group $G = O(2)$. Indeed, on low frequency datasets, the bound tends to saturate faster as we increase the group size $|H|$. Conversely, on high frequency datasets, choosing a larger group $H$ results in larger improvements in the estimated bound.

*Remark* 5.1. As it can be seen in our experiments in Fig 2 and 3, our bound is larger that the one in Neyshabur et al. [2018]. This is partially due to coarse approximation of Frobenius norm and the presence of the term $h^2$ in our bound. This dependency could be alleviated if we increase the number of used irreps in the group convolutional network, namely the larger group size. It is an interesting research direction to explore the optimal dependency on the width for equivariant networks.

In Fig. 4, we perform a larger study on the transformed MNIST datasets to investigate the simultaneous effect of the group size $|H|$, data augmentation and training set size $m$. For each training data size, the generalization error improves with larger equivariance group $H$.

We study the relation between the term $\sum_l \sqrt{M(l, \eta)}$ and the generalization error observed. Note that this term depends only on the architecture design, i.e. it is data and training agnostic. In Fig. 5, we show the correlation between these two quantities for two synthetic datasets. Both datasets have $G = O(2)$ symmetry, but the data rotates with frequencies up to $F = 1$ in the first case and $F = 6$ in the second. We observe that the term $\sum_l \sqrt{M(l, \eta)}$ correlates strongly with the generalization in the high-frequency dataset ($F = 6$), where the generalization benefits from increasing the size of the equivariance group. Conversely, in the low-frequency case ($F = 1$), we observe a saturation of the performance, as increasing the group size does not improve generalization. In this setting, the term $\sum_l \sqrt{M(l, \eta)}$ is only a weak predictor of generalization. From these observations, we interpret this term as an indicator of the expressiveness of a model. Particularly, when the irreps are chosen compatible with the dataset symmetries, the term $\sum_l \sqrt{M(l, \eta)}$ provides a good guideline for the architecture choice. However, similar to general neural networks, the generalization error is dependent on other factors as well, which can be for example norms of network weights.

# 6   Conclusion

We provided learning theoretic frameworks to study the effect of invariance and equivariance on the generalization error and provide guidelines for design choices. The obtained bounds provide

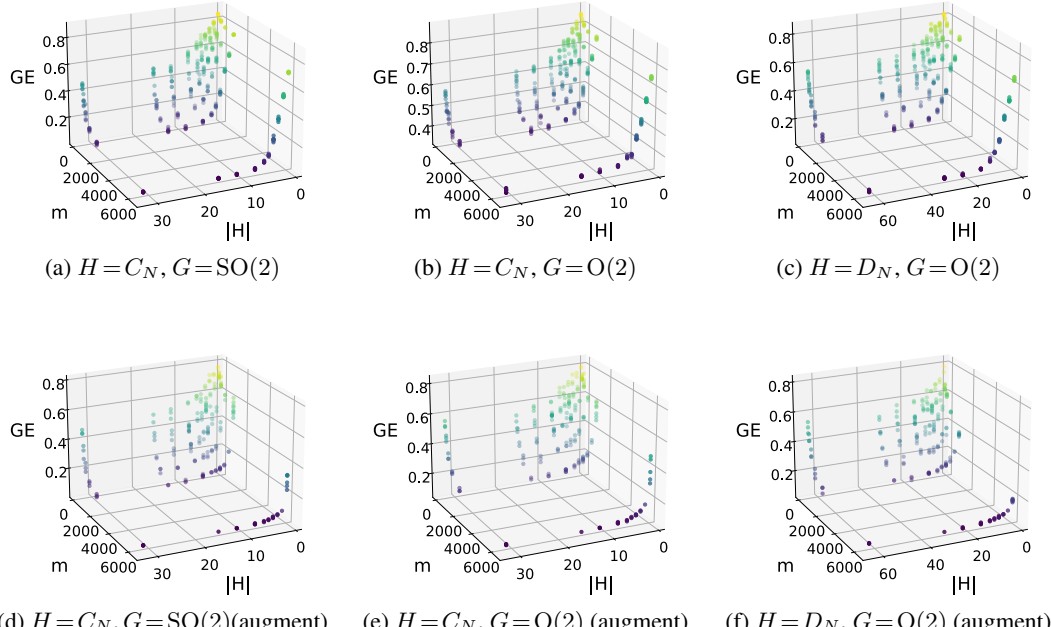

(a) $H = C_N, G = SO(2)$
(b) $H = C_N, G = O(2)$
(c) $H = D_N, G = O(2)$

(d) $H = C_N, G = SO(2)$(augment)
(e) $H = C_N, G = O(2)$ (augment)
(f) $H = D_N, G = O(2)$ (augment)

Figure 4: Generalization Error (GE) of different $H$-equivariant models ($H = C_N$ or $H = D_N$) on different $G$-MNIST datasets ($G = SO(2)$ or $G = O(2)$) for different training set sizes $m$.

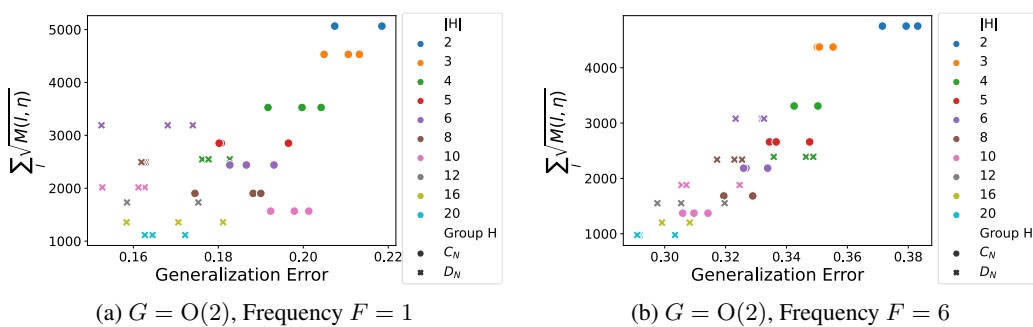

(a) $G = O(2)$, Frequency $F = 1$
(b) $G = O(2)$, Frequency $F = 6$

Figure 5: Correlation between the term $\sum_l \sqrt{M(l, \eta)}$ and the Generalization Error (GE) of different $H$-equivariant models ($H = C_N$ or $H = D_N$) on the synthetic $O(2)$ dataset with frequency $F = 1$ and $F = 6$, with a training set size $m = 3200$.

useful insights and guidelines for understanding these architectures. As limitation of our work, we show experimentally in supplementary materials that the obtained bounds do not decrease with larger training sizes $m$ as expected. Nevertheless, for a fixed training set, the trend of the generalization bound correlates with the generalization error.

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
