## Supplementary Material

## 7 Elements of Group and Representation Theory

In this section, we provide a brief introduction to the concepts from Group Theory which we need in our derivations.

**Definition 7.1** (Group). A *group* is a pair $(G, \cdot)$ containing a set $G$ and a binary operation $\cdot : G \times G \to G, (h, g) \mapsto h \cdot g$ which satisfies the group axioms:

- Associativity:    $\forall a, b, c \in G \quad a \cdot (b \cdot c) = (a \cdot b) \cdot c$

- Identity:    $\exists e \in G : \forall g \in G \quad g \cdot e = e \cdot g = g$

- Inverse:    $\forall g \in G \ \exists g^{-1} \in G : \quad g \cdot g^{-1} = g^{-1} \cdot g = e$

The operation $\cdot$ is the *group law* of $G$.

The inverse elements $g^{-1}$ of an element $g$, and the identity element $e$ are *unique*. In addition, if the group law is also *commutative*, the group $G$ is an *abelian group*. To simplify the notation, we commonly write $ab$ instead of $a \cdot b$. It is also common to denote the group $(G, \cdot)$ just with the name of its underlying set $G$.

The **order** of a group $G$ is the *cardinality* of its set and is indicated by $|G|$. A group $G$ is **finite** when $|G| \in \mathbb{N}$, i.e., when it has a finite number of elements. A **compact** group is a group that is also a compact topological space with continuous group operation.

**Definition 7.2** (Group Action). Given a group $G$, its *action* on a set $\mathcal{X}$ is a map $. : G \times \mathcal{X} \to \mathcal{X}, (g, x) \mapsto g.x$ which satisfies the axioms:

- identity: $\forall x \in \mathcal{X} \quad e.x = x$

- compatibility: $\forall a, b \in G \ \forall x \in \mathcal{X} \quad a.(b.x) = (ab).x$

A simple example of group action is the group law itself $\cdot : G \times G \to G$ which defines an action of $G$ on its own elements ($\mathcal{X} = G$). Another important action is the one defined on signals overs the group $G$. Given a signal $x : G \to \mathbb{R}$, the action of an element $g \in G$ maps $x \mapsto g.x$, $[g.x](h) := x(g^{-1}h)$.

The **orbit** of $x \in \mathcal{X}$ through $G$ is the set $G.x := \{g.x | g \in G\}$. The orbits of the elements in $\mathcal{X}$ form a *partition* of $\mathcal{X}$. By considering the equivalence relation $\forall x, y \in \mathcal{X} \quad x \sim_G y \iff x \in G.y$ (or, equivalently, $y \in G.x$), one can define the **quotient space** $\mathcal{X}/G := \{G.x | x \in \mathcal{X}\}$, i.e. the set of all different orbits.

**Definition 7.3** (Subgroup). Given a group $(G, \cdot)$, a non-empty subset $H \subseteq G$ is a **subgroup** of $G$ if it forms a group $(H, \cdot)$ under the same group law, restricted to the elements of $H$. This is usually denoted as $H \leq G$.

$H$ is a subgroup of $G$ if and only if the subset $H$ is *closed* under the group law and the inverse operations, i.e.:

- $\forall a, b \in H \quad a \cdot b \in H$
- $\forall h \in H \quad h^{-1} \in H$

Note that a subgroup $H$ needs to contain the identity element $e \in G$.

**Linear representations**

**Definition 7.4** (Linear representation). Given a group $G$ and a vector space $V$, a linear representation of $G$ is a homomorphism $\rho : G \to \mathrm{GL}(V)$ which associates to each element of $g \in G$ an element of general linear group $\mathrm{GL}(V)$, i.e. an invertible matrix acting on $V$, such that the condition below is satisfied:

$$\forall g, h \in G, \quad \rho(gh) = \rho(g)\rho(h).$$

The most simple representation is the trivial representation $\psi : G \to \mathbb{R}, g \mapsto 1$, mapping all element to the multiplicative identity $1 \in \mathbb{R}$. The common 2-dimensional rotation matrices are an example of representation on $\mathbb{R}^2$ of the group $\mathrm{SO}(2)$.

For a finite group $G$, an important representation is the *regular representation* $\rho_{\mathrm{reg}}$. It acts on the space $V = \mathbb{R}^{|G|}$ of vectors representing signals over the group $G$. The regular representation $\rho_{\mathrm{reg}}(g)$ of an element $g \in G$ is a $|G| \times |G|$ matrix which permutes the entries of vectors $\boldsymbol{x} \in \mathbb{R}^{|G|}$. Each vector in $\mathbb{R}^{|G|}$ represents a function over the group, $x : G \to \mathbb{R}$, with $x(g_i)$ being $i$-th entries of $\boldsymbol{x}$. Then, the group action $\rho_{\mathrm{reg}}(g)\boldsymbol{x}$ represents the function $g.x$, i.e. the signal $x$ shifted by $g$. In other words, $\rho_{\mathrm{reg}}(g)$ is a permutation matrix moving the $i$-th entry of $\boldsymbol{x}$ to the $j$-th entry such that $g_j = gg_i$. Regular representations are of high importance, because they describe the features of group convolution networks.

Given two representations $\rho_1 : G \to \mathrm{GL}(\mathbb{R}^{n_1})$ and $\rho_2 : G \to \mathrm{GL}(\mathbb{R}^{n_2})$, their *direct sum* $\rho_1 \oplus \rho_2 : G \to \mathrm{GL}(\mathbb{R}^{n_1+n_2})$ is a representation obtained by stacking the two representations as follow:

$$(\rho_1 \oplus \rho_2)(g) = \begin{bmatrix} \rho_1(g) & 0 \\ 0 & \rho_2(g) \end{bmatrix} .$$

Note that this representation acts on $\mathbb{R}^{n_1+n_2}$ which contains the concatenation of the vectors in $\mathbb{R}^{n_1}$ and $\mathbb{R}^{n_2}$.

**Fourier Transform**   Fourier analysis can be generalized for square integrable complex functions $f : G \to \mathbb{C}$ defined over a compact group $G$. Fourier components in this case are a particular set of complex representations of $G$ called *irreducible representations* (or *irreps*) of $G$. Irreps are defined as representation with no $G-$invariant subspaces (see Serre [1977] for rigorous details). The key role of irreps is clear from the following theorem.

**Theorem 7.5** (Maschke's Theorem). *Every representation of a finite group $G$ on a nonzero, finite dimensional complex vector space is a direct sum of irreducible representations.*

Maschke's theorem can be generalized to compact groups. Irreps are, therefore, simple blocks for decomposing representations. Irreps are denoted by $\psi$ with dimension $\dim_\psi$, namely, $\psi : G \to \mathrm{GL}(\mathbb{C}^{\dim_\psi})\}_\psi$. The set of irreps is denoted by $\hat{G} = \{\psi : G \to \mathrm{GL}(\mathbb{C}^{\dim_\psi})\}$. The matrix coefficients of the irreps (i.e. each entry $\psi_{ij} : G \to \mathbb{C}$ of each irrep $\psi$, interpreted as a function over $G$) form an orthogonal basis for square integrable functions over $G$. Therefore, we can project a function $f : G \to \mathbb{C}$ on this basis and get the Fourier components as:

$$\hat{f}(\psi) := \int_{g \in G} f(g)\psi(g) \, d\mu(g),$$

where $\mu$ is the Haar measure over $G$. Note that $\hat{f}(\psi)$ is a $\dim_\psi \times \dim_\psi$ matrix containing a coefficient for each entry of $\psi$. The inverse Fourier transform is defined as:

$$f(g) = \sum_{\psi \in \hat{G}} \dim_\psi \mathrm{Tr}(\hat{f}(\psi)\psi(g^{-1}))$$

Note that the trace is equivalent to computing the elementwise product between the matrices $\hat{f}(\psi)$ and $\psi(g^{-1})^T = \overline{\psi(g)}$ and then summing over all entries.

If $G$ is the cyclic group, $G = \mathrm{C}_N$, one recovers the usual *discrete Fourier transform*, because the complex irrep of $\mathrm{C}_N$ are 1-dimensional and correspond to the the complex exponentials up to order $N - 1$. When $G$ is finite, the number of Fourier coefficients $\sum_{\psi \in \hat{G}} \dim_\psi^2$ is equal to $|G|$. Indeed, the Fourier transform of a signal $x : G \to \mathbb{C}$ can be interpreted as a $|G| \times |G|$ change of basis which maps the vector $\boldsymbol{x} \in \mathbb{C}^{|G|}$ representing $x$ in the group domain to $\hat{\boldsymbol{x}} \in \mathbb{C}^{|G|}$ containing all the Fourier coefficients.

Given two signals $w, x : G \to \mathbb{C}$, the following property (similar to the common Fourier transform) holds:

$$\widehat{g.x}(\psi) = \psi(g)\hat{x}(\psi) \tag{12}$$

Eq. 12 guarantees that a transformation by $g$ of $x$ does not mix the coefficients associated to different irreps. Moreover, the coefficients associated with the irrep $\psi$ are mixed precisely by the matrix $\psi(g)$, i.e. they transform according to $\psi$. Indeed, if all the coefficients associated to the same irrep are grouped together, one can decompose the regular representation $\rho_{\text{reg}}$ of $G$ in a *direct sum* of the irreps of $G$, up to the change of basis $B$ as above. More precisely, the irreps decomposition of $\rho_{\text{reg}}$ contains $\dim_\psi$ copies of $\psi$, one for each column[5] of $\hat{f}(\psi)$, i.e.:

$$\rho_{\text{reg}}(g) = B^{-1} \bigoplus_{\psi \in \hat{G}} \left( \underbrace{\psi(g) \oplus \psi(g) \oplus \dots}_{\dim_\psi \text{ times}} \right) B \ .$$

This is the key insight used in the analysis of group convolution in this work.

**Group Convolution**    Given a space $\mathcal{X}$ associated with the action of a group $G$ and an invariant inner product $\langle \cdot, \cdot \rangle : \mathcal{X} \times \mathcal{X} \to \mathbb{R}$, we define the *group convolution* $w \circledast_G x : G \to \mathbb{R}$ of two elements $w, x \in \mathcal{X}$ as

$$\forall g \in G \quad (w \circledast_G x)(g) := \langle g.w, x \rangle \ . \tag{13}$$

Group convolution satisfies the following equality:

$$\widehat{w \circledast_G x}(\psi) = \hat{x}(\psi) \hat{w}(\psi)^T \tag{14}$$

The classical definition of *group convolution* between two signals $w, x : G \to \mathbb{R}$ is a special case of the one above in the case $\mathcal{X}$ is the space of square integrable functions over $G$ and the invariant inner product is defined as:

$$\langle w, x \rangle := \int_{g \in G} w(g) x(g) \, d\mu(g)$$

where $\mu : G \to \mathbb{R}$ is a Haar measure over $G$. The group convolution, then, becomes

$$(w \circledast_G x)(g) := \int_{h \in G} w(g^{-1}h) x(h) \, d\mu(h) = \int_{h \in G} w(h) x(g.h) \, d\mu(h) \ .$$

Note that what we defined is technically a **group cross-correlation**, and so it differs from the usual definitions of convolution over groups. We still refer to is as group convolution to follow the common terminology in the deep learning literature.

When $G$ is finite, the Haar measure is the counting measure and the integral becomes a sum. Fix an ordering of group elements as $(g_0, g_1, \dots, g_i, \dots, g_{|G|-1})$ with $g_0 = e$ the identity element. The signals $x, w : G \to \mathbb{R}$ can be stored as $|G|$ dimensional vectors $\boldsymbol{x}, \boldsymbol{w} \in \mathbb{R}^{|G|}$, where the $i$-th entry contains the value of the function evaluated on $g_i$:

$$\boldsymbol{x} = \begin{pmatrix} x(g_0) \\ \vdots \\ x(g_{|G|-1}) \end{pmatrix}, \quad \boldsymbol{w} = \begin{pmatrix} w(g_0) \\ \vdots \\ w(g_{|G|-1}) \end{pmatrix} . \tag{15}$$

The same holds for the output signal $w \circledast_G x$. Define matrix $\boldsymbol{W}$ as follows:

$$\boldsymbol{W} = \begin{pmatrix} w(g_0) & \cdots & w(g_{|G|-1}) \\ w(g_1^{-1}g_0) & \cdots & w(g_1^{-1}g_{|G|-1}) \\ \vdots & \ddots & \vdots \\ w(g_{|G|-1}^{-1}g_0) & \cdots & w(g_{|G|-1}^{-1}g_{|G|-1}) \end{pmatrix} \tag{16}$$

Then, the group convolution can then be expressed as a matrix multiplication between the vector $\boldsymbol{x}$ and a matrix $\boldsymbol{W}$ containing at position $(i, j)$ the entry of $w_{g_i^{-1}g_j}$:

$$(w \circledast_G h.x)(g_i) = (\boldsymbol{W}\boldsymbol{x})[i] \tag{17}$$

The matrix $\boldsymbol{W}$ contains permuted copies of its first row, where the respective permutation matrix is determined by group actions. Assuming $g_0 = e$ is the identity, the first row of $\boldsymbol{W}$ contains $\boldsymbol{w}^T$ while the $i$-th row contains $(g_i.\boldsymbol{w})^T$, i.e. the vector representing the signal $g_i.w$. The matrix $\boldsymbol{W}$ is a **$G$-circulant matrix**.

---

[5]In Eq. 12, $\psi(g)$ acts on each column of the matrix $\hat{f}(\psi)$ independently.

**Equivariance.** For two vector spaces $V$ and $V'$ with group representations respectively $\rho$ and $\rho'$, the linear transformation $T : V' \to V$ is equivariant if

$$\forall g \in G : T \circ \rho'(g) = \rho(g) \circ T.$$

Now, since an invariant inner product satisfies $\langle g.w, g.x \rangle = \langle w, x \rangle$ for any element $g \in G$, one can show that the group convolution above is **equivariant**:

$$\begin{aligned}
(w \circledast_G h.x)(g) &= \langle g.w, h.x \rangle \\
&= \langle (h^{-1}g).w, (h^{-1}h).x \rangle \\
&= \langle (h^{-1}g).w, x \rangle \\
&= (w \circledast_G x)(h^{-1}g) \,.
\end{aligned}$$

By definition of action of $G$ on signals over $G$, it follows that $(w \circledast_G h.x) = h.(w \circledast_G x)$.

We finish the part by presenting Schur's lemma, [[Serre, 1977, Section 2.2]] which helps characterizing equivariant maps.

**Theorem 7.6** (Schur's lemma). *Let $\rho^1$ and $\rho^2$ be irreducible representations of $G$ respectively on vector spaces $V_1$ and $V_2$. Suppose that the linear transformation $T : V_1 \to V_2$ is equivariant. Then:*

1. *If $\rho^1$ and $\rho^2$ are non-isomorphic, then $T = 0$.*

2. *If $V_1 = V_2$ and $\rho^1 = \rho^2$, then $T$ is a homothety, i.e, a scalar multiple of identity.*

**Equivariant neural networks.** To build equivariant networks, without loss of generality, we assume the feature $\boldsymbol{x} \in \mathbb{R}^{c_l}$ of the neural network at layer $l$ transforms according to a generic representation $c_l$-dimensional representation $\rho_l$ and that it is decomposed[6] in a number of subvectors $\hat{\boldsymbol{x}}_i \in \mathbb{R}^{c_{l,i}}$, with $\sum_i c_{l,i} = c_l$, each transforming according to an irrep $\psi_i : G \to \mathrm{GL}(\mathbb{C}^{c_{l,i}})$. To see this more precisely, consider the layer $l$ given by the matrix $\boldsymbol{W}_l \in \mathbb{R}^{c_l \times c_{l-1}}$. The group representation is $\rho_l$ acting on $\mathbb{R}^{c_l}$. The equivariance relation is given by:

$$\boldsymbol{W}_l \rho_{l-1}(h) = \rho_l(h) \boldsymbol{W}_l \quad .$$

We first decompose representations using Maschke's theorem. The representation $\rho_l$ can be written direct sum of irreps as

$$\rho_l = Q_l \left( \bigoplus_{\psi} \bigoplus_{i=1}^{m_{l,\psi}} \psi \right) Q_l^{-1}$$

$$= Q_l \times \text{block-diagonal} \left( \psi \in \hat{G} : \underbrace{\begin{pmatrix} \psi & \cdots & 0 \\ \vdots & \ddots & \vdots \\ 0 & \cdots & \psi \end{pmatrix}}_{m_{l,\psi} \text{ blocks}} \right) \times Q_l^{-1}$$

The multiplicity of the irrep $\psi$ in $\rho_l$ is $m_{l,\psi}$. Using $\widehat{\boldsymbol{W}}_l = Q_l^{-1} \boldsymbol{W}_l Q_{l-1}$, we have

$$\widehat{\boldsymbol{W}}_l \left( \bigoplus_{\psi} \bigoplus_{i=1}^{m_{l-1,\psi}} \psi \right) = \left( \bigoplus_{\psi} \bigoplus_{i=1}^{m_{l,\psi}} \psi \right) \widehat{\boldsymbol{W}}_l.$$

Since $\left( \bigoplus_{\psi} \bigoplus_{i=1}^{m_{l,\psi}} \psi \right)$ is block diagonal, we can partition $\widehat{\boldsymbol{W}}_l$ similarly with the sub-block $(j, i)$ of blocks relating $\psi_1$ to $\psi_2$, denoted by $\widehat{\boldsymbol{W}}_l(\psi_2, j, \psi_1, i)$. The block relates $\psi_1$ to $\psi_2$ as:

$$\widehat{\boldsymbol{W}}_l(\psi_2, j, \psi_1, i)\psi_1 = \psi_2 \widehat{\boldsymbol{W}}_l(\psi_2, j, \psi_1, i).$$

---

[6]The decomposition can include an orthogonal change of basis $Q$.

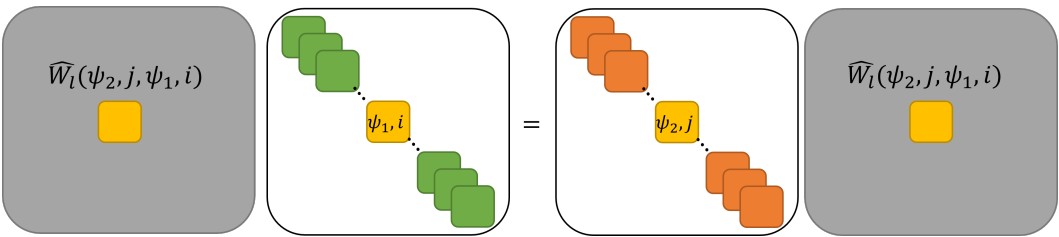

Figure 6: The block diagonal structure in $\boldsymbol{W}_l \rho_{l-1}(h) = \rho_l(h)\boldsymbol{W}_l$

This setup follows the premise of Schur's lemma, exept that in practice, a neural network uses real valued features and weights. Hence, we need to consider *real* irreps, which we denote with $\{\psi_i\}$. This requires some adaptation to the Fourier analysis and Schur's lemma. This is explained more precisely in Supplementary 8. If $\psi_1 \neq \psi_2$, the block needs to be zero. Otherwise, each block $\boldsymbol{W}$ equivariant to irrep $\psi$ can be expressed as

$$\boldsymbol{W} = \sum_{k=1}^{c_\psi} w_k \boldsymbol{B}_{\psi,k}$$

where matrices $\boldsymbol{B}_{\psi,k}$ are given in Supplementary 8, $c_\psi$ is either 1,2 or 4, and $\{w_k\}_{k=1}^{c_\psi}$ are the $c_\psi$ free parameters. Therefore, each block $\widehat{\boldsymbol{W}}_l(\psi, j, \psi, i)$ is characterized by $c_\psi$ free parameters.

Note that, the main difference of real irreps is that only the matrix coefficients in a fraction $\frac{1}{c_\psi}$ of the columns of $\psi$ are part of the orthogonal basis for real functions on $G$, while the other coefficients are redundant. It follows that, when computing the Fourier transform of a function $x : G \to \mathbb{R}$, only the first $\frac{\dim_\psi}{c_\psi}$ columns of $\hat{x}(\psi)$ needs to be considered and, therefore, the regular representation of $H$ only contains $\frac{\dim_\psi}{c_\psi}$ copies of $\psi$.

**Group convolutional networks.** Group convolutional networks is an important special case of the setup presented above. Group convolution based architectures of Cohen and Welling [2016a] can be obtained by choosing multiple copies of regular representations at each layer.

Consider an input vector space $\mathcal{X}$ associated with an action $H \times \mathcal{X} \to \mathcal{X}$ of $H$. We use group convolution as defined in eq. 13. Each convolutional kernel can be parametrized by a kernel $\boldsymbol{w}$, and its action is represented by a group circulant matrix $\boldsymbol{W}$ as in eq. 16.

This is the building blocks of the linear layers of group equivariant networks. Each intermediate convolutional layer maps a feature space with $c_{l-1}$ channels of dimension equal to group size $|H|$ to a new feature space with $c_l$ channels of dimension $|H|$. Any such layer can be visualized as follows:

$$\begin{pmatrix} \mathbf{y}_1 \in \mathbb{R}^{|H|} \\ \mathbf{y}_2 \in \mathbb{R}^{|H|} \\ \vdots \\ \mathbf{y}_{c_l} \in \mathbb{R}^{|H|} \end{pmatrix} = \underbrace{\begin{pmatrix} \boldsymbol{W}_{l,(1,1)} & \boldsymbol{W}_{l,(2,1)} & \cdots & \boldsymbol{W}_{l,(c_{l-1},1)} \\ \boldsymbol{W}_{l,(1,2)} & \boldsymbol{W}_{l,(2,2)} & \cdots & \boldsymbol{W}_{l,(c_{l-1},2)} \\ \vdots & \ddots & & \vdots \\ \boldsymbol{W}_{l,(1,c_l)} & \boldsymbol{W}_{l,(2,c_l)} & \cdots & \boldsymbol{W}_{l,(c_{l-1},c_l)} \end{pmatrix}}_{\boldsymbol{W}_l} \cdot \underbrace{\begin{pmatrix} \mathbf{x}_1 \in \mathbb{R}^{|H|} \\ \mathbf{x}_2 \in \mathbb{R}^{|H|} \\ \vdots \\ \mathbf{x}_{c_{l-1}} \in \mathbb{R}^{|H|} \end{pmatrix}}_{\boldsymbol{x}} \qquad (18)$$

Here, any $\boldsymbol{W}_{l,(i,j)} \in \mathbb{R}^{|H| \times |H|}$ block an group circulant matrix corresponding to $H$, which encodes a $H$-convolution parametrised by a filter $\boldsymbol{w}_{l,i,j} \in \mathbb{R}^{|H|}$. Note that the output of the $l$-th layer is seen as a $|H| \times c_l$ signal, i.e. it stores a different value $(\boldsymbol{w}_1 \circledast_H \boldsymbol{x})(h, i)$ for each group element $h$ and channel $i$.

Usually, the group action on the input space $\mathcal{X}$ is given by the task. The first linear layer transforms the input space to multiple signals (channels) over $H$. From that moment, we can use the convolution in Eq. 13 in the following layers.

Finally, the last layer produces an invariant output for each class. This can be interpreted as a group convolution with constant filters or, equivalently, as an average pooling within the $|H|$ channels in each block to produce a $c_{L-1}$ dimensional invariant vector followed by a linear classifier.

The resulting architecture can be written as a MLP with structured weights similar to Wood and Shawe-Taylor [1996], Ravanbakhsh [2020]. Such network, denoted by $f(\boldsymbol{x}; \mathbb{W})$, is an MLP with $L$ layers, input $\boldsymbol{x}$ and the set of effective parameters $\mathbb{W}$. The $l$-th layer consists of a linear map $\boldsymbol{W}_l$ from a vector space $\mathcal{V}_{l-1}$ to another vector space $\mathcal{V}_l$ followed by an activation function $\sigma_l(\cdot)$. The network output at the layer $l$, denoted by $f_l(\boldsymbol{x}; \mathbb{W})$, is given by $f_l(\boldsymbol{x}; \mathbb{W}) = \sigma_l(\boldsymbol{W}_l f_{l-1}(\boldsymbol{x}; \mathbb{W}))$.

This design guarantees that each linear layer commutes with the group action. If also all non-linearities commute with it, i.e.

$$\sigma_l(\boldsymbol{W}_l h.\boldsymbol{x}) = \sigma_l(h.(\boldsymbol{W}_l \boldsymbol{x})) = h.\sigma_l(\boldsymbol{W}_l \boldsymbol{x}) ,$$

it follows that the output of the model is invariant to $H$.

Note that the number of *effective channels* after the $l$-th hidden layer is $|H| \, c_l$ but its number of trainable parameters is $|H| \, c_l c_{l-1}$. In total, if number of channels across layers are all equal to c, then the number trainable parameters is equal to $\mathcal{O}\left(L|H|c^2\right)$. In our experiments, we always consider fixed architectures where the number of *effective channels* is kept constant when changing the group, i.e. different group choices only affect the number of learnable parameters and the structure of the weight matrices but not their size and, therefore, not the computational cost of the model.

**Groups considered in this work.** In the experiments, we consider 4 different families of compact groups. The special orthogonal group $\mathrm{SO}(2)$ is the group of all planar rotations. It is an abelian (commutative) group and contains any rotation by an angle $\theta \in [0, 2\pi)$. $\mathrm{SO}(2)$ contains infinite elements. The orthogonal group $\mathrm{O}(2)$ is the group of all planar rotations and reflections. Indeed, $\mathrm{SO}(2)$ is a subgroup of $\mathrm{O}(2)$. The elements of $\mathrm{O}(2)$ are planar rotations, or planar rotations combined with a reflection along a fixed axis. It is not commutative and contains infinite elements. The cyclic group $\mathrm{C}_N$ of order $N$ is a finite group containing the $N$ rotations by angles multiple of $\frac{2\pi}{N}$. It is a subgroup of both $\mathrm{SO}(2)$ and $\mathrm{O}(2)$ and it is abelian. Finally, the dihedral group $\mathrm{D}_N$ is a finite group containing $2N$ elements. It contains the $N$ rotations in $\mathrm{C}_N$ and but also their composition with a reflection along a fixed axis. It is a subgroup of $\mathrm{O}(2)$ and it is not commutative. If $G = \mathrm{C}_N$, the matrix $\boldsymbol{W}$ is a circulant matrix in the classical sense, where each row is a cyclic shift by one step of the previous one.

## 8   Schur's Lemma for intertwiners between real representations

Given two *complex* irreps $\sigma_i : G \to \mathrm{GL}(\mathbb{C}^{\dim \psi_i})$ and $\sigma_j : G \to \mathrm{GL}(\mathbb{C}^{\dim \sigma_j})$, Schur's Lemma guarantees that the set of equivariant matrices (**intertwiners**)

$$\mathrm{Hom}_{G, \mathbb{C}}(\psi_i, \psi_j) := \{M \mid M\psi_i(g) = \psi_j(g)M \quad \forall g \in G\}$$

is either zero dimensional containing only zero function (if $\psi_i \not\cong \psi_j$), or 1-dimensional and contains only scalar multiples of the identity (if $\psi_i = \psi_j$), i.e.

$$\forall g \in G \quad M\psi(g) = \psi(g)M \iff \exists \lambda \in \mathbb{C} \text{ s.t. } M = \lambda I,$$

where $\cong$ denotes isomorphism. In other words, there is a change of basis matrix such that the two representations are equal. The space $\mathrm{Hom}_{G, \mathbb{C}}(\psi_i, \psi_j)$ is called the space of homomorphisms. The subscript $\mathbb{C}$ indicates the complex-valued intertwiners are intended. This lemma is the core of equivariant (complex valued) linear layers and provides a way to parametrize the space of all such operations.

In this section, we discuss Schur's lemma for real irreps and characterize the space of homomorphisms $\mathrm{Hom}_{G, \mathbb{R}}(\psi_i, \psi_j)$ for real valued representations. We still use $\sigma$ to denote *complex irreps* and will refer to *real irreps* with $\psi$. First we note that if $\psi_i \not\cong \psi_j$ are different *real* irreps, by using the complex version of Schur's Lemma, one can show that $\mathrm{Hom}_{G, \mathbb{R}}(\psi_i, \psi_j)$ still contains only the zero matrix.

When we have $\psi_i \cong \psi_j$, the derivations are more tricky. For this cases, $\psi_i = \psi_j = \psi$, we have to consider $\mathrm{Hom}_{G, \mathbb{R}}(\psi, \psi)$. For complex-valued representations, as seen above from Schur's lemma, this space is one dimensional spanned by the identity matrix $\boldsymbol{I}$. However, for real representations, this space can be either 1, 2 or 4 dimensional, corresponding to three types: real $\mathbb{R}$, complex $\mathbb{C}$ and quanternion $\mathbb{H}$ (see [Cesa et al., 2022, Section C] for accessible explanations). Based on this, any real irrep $\psi$ can be classified in three categories:

- **real-type**: $\exists \psi$ real $: \psi \cong \sigma \cong \overline{\sigma}$

- **complex-type**: $\exists \psi$ real $: \psi = \sigma_{\mathbb{R}} \cong \sigma \oplus \overline{\sigma}$, with $\sigma \not\cong \overline{\sigma}$

- **quaternionic-type**: $\exists \psi$ real $: \psi = \sigma_{\mathbb{R}} \cong \sigma \oplus \overline{\sigma} \cong \sigma \oplus \sigma$, as $\sigma \cong \overline{\sigma}$

where $\overline{\sigma}$ is a representation such that $\overline{\sigma} : g \mapsto \overline{\sigma(g)}$ and $\overline{\sigma(g)}$ is the complex-conjugation of the entries of the matrix $\sigma(g)$. We also have defined

$$\sigma_{\mathbb{R}}(g) := \begin{bmatrix} Re\left(\sigma(g)\right) & -Im\left(\sigma(g)\right) \\ Im\left(\sigma(g)\right) & Re\left(\sigma(g)\right) \end{bmatrix} \in \mathbb{R}^{2d \times 2d}$$

where $Re\left(X\right)$ (respectively, $Im\left(X\right)$) is a matrix containing the *real* (respectively, *imaginary*) part of the complex entries of the matrix $X$, i.e.:

$$Re\left(X\right) = \frac{1}{2}\left(X + lineX\right)$$

$$Im\left(X\right) = -i\frac{1}{2}\left(X - \overline{X}\right)$$

$$X = Re\left(X\right) + iIm\left(X\right)$$

Using these definitions, one can also verify that for any $d$-dimensional complex irrep $\sigma$:

$$\sigma_{\mathbb{R}}(g) = D_d \begin{bmatrix} \sigma(g) & 0 \\ 0 & \overline{\sigma}(g) \end{bmatrix} D_d^* = D_d(\sigma(g) \oplus \overline{\sigma}(g))D_d^*$$

where

$$D_d = \frac{1}{\sqrt{2}} \begin{bmatrix} iI_d & -iI_d \\ I_d & I_d \end{bmatrix}$$

where $I_d$ is the $d \times d$ identity matrix and $*$ indicates the conjugate transpose. Note that $D_d$ is a unitary matrix.

We can then distinguish three cases.

**Real type**    If $\psi \cong \sigma \cong \overline{\sigma}$, we have $\psi = A\sigma A^*$, with $A$ unitary matrix $(A^{-1} = A^*)$. An intertwiner matrix $M$ satisfies $M\psi(g) = \psi(g)M$. It can be seen that the matrix $\tilde{M} := A^*MA$, $\tilde{M}$ is an intertwiner of $\sigma$:

$$\tilde{M}\sigma(g) = A^*MA\sigma(g) = A^*M\psi(g)A = A^*\psi(g)MA = \sigma(g)A^*MA = \sigma(g)\tilde{M},$$

and, therefore, has form $\lambda I$ for $\lambda \in \mathbb{C}$ from complex-valued Schur's lemma. It follows that $M = \lambda I$, allowing only real intertwiners,

$$M_{\mathbb{R}} = \lambda I$$

with $\lambda \in \mathbb{R}$.

**Complex type**    If $\psi \cong \sigma \oplus \overline{\sigma}$, we have $\psi = D_d(\sigma \oplus \overline{\sigma})D_d^*$ with $d$ being the dimension of irrep $\sigma$. If $M$ satisfies $M\psi(g) = \psi(g)M$, then the matrix $\tilde{M} = D_d^*MD_d$ is an intertwiner of $\sigma \oplus \overline{\sigma}$:

$$\begin{aligned} \tilde{M}(\sigma(g) \oplus \overline{\sigma}(g)) &= D_d^*MD_d(\sigma(g) \oplus \overline{\sigma}(g)) \\ &= D_d^*M\psi D_d \\ &= D_d^*MD_d(\sigma(g) \oplus \overline{\sigma}(g)) = (\sigma(g) \oplus \overline{\sigma}(g))\tilde{M}. \end{aligned}$$

We can apply complex-valued Schur's lemma again. The matrix $\tilde{M}$ needs to have form $\alpha I \oplus \beta I$ for $\alpha, \beta \in \mathbb{C}$. Re-applying the change of basis $D_d$ and requiring only real entries, one finds that $M$ needs to have the following form:

$$M_{\mathbb{C}} = \begin{bmatrix} aI_d & -bI_d \\ bI_d & aI_d \end{bmatrix}$$

with $a, b \in \mathbb{R}$.

**Quaternionic type** If $\psi \cong \sigma \oplus \sigma$, there is a basis where $\psi = D_d(\sigma \oplus \overline{\sigma})D_d^*$ and $\overline{\sigma} = T_d^T \sigma T_d$ with $T_d = \begin{bmatrix} 0 & -I_{d/2} \\ I_{d/2} & 0 \end{bmatrix}$, that is $\psi = D_d(I_d \oplus T_d^T)(\sigma \oplus \sigma)(I_d \oplus T_d)D_d^*$. If $M$ satisfies $M\psi(g) = \psi(g)M$, we define $\tilde{M} = D_d^*(I_d \oplus T_d)M(I_d \oplus T_d^T)D_d$, intertwiner of $\sigma \oplus \sigma$. $\tilde{M}$ needs to have form $\tilde{M} = \begin{bmatrix} \alpha I_d & \gamma I_d \\ \beta I_d & \delta I_d \end{bmatrix}$ for $\alpha, \beta, \gamma, \delta \in \mathbb{C}$. Re-applying the change of basis $D_d(I_d \oplus T_d^T)$ and requiring only real entries, one can show that $M$ needs to have the following form:

$$M_\mathbb{H} = \begin{bmatrix} aI_d + cT_d & -bI_d + dT_d \\ bI_d + dT_d & aI_d - cT_d \end{bmatrix} = \begin{bmatrix} aI_d & -cI_d & -bI_d & -dI_d \\ cI_d & aI_d & dI_d & -bI_d \\ bI_d & -dI_d & aI_d & cI_d \\ dI_d & bI_d & -cI_d & aI_d \end{bmatrix} = \begin{bmatrix} a & -c & -b & -d \\ c & a & d & -b \\ b & -d & a & c \\ d & b & -c & a \end{bmatrix} \otimes I_d$$

with $a, b, c, d \in \mathbb{R}$.

**Application to equivariant networks.** Real-valued version of Schur's lemma characterizes equivariant kernel $W$ parametrized by $M_\mathbb{R}, M_\mathbb{C}, M_\mathbb{H}$. The columns of $W$ are all orthogonal to each other and each contains only one copy of each of its $c_\psi$ free parameters, where

$$c_\psi = \begin{cases} 1 & \text{real type} \\ 2 & \text{complex type} \\ 4 & \text{quaternionic type} \end{cases}$$

It follows that all columns have the same norm, equal to the sum of the $c_\psi$ free parameters squared, and therefore that $W$ is a scalar multiple of an orthonormal matrix (with scale equal to the norm of any of its columns). The matrix $W$ satisfies the following property:

$$\|Wx\|_2^2 = \frac{1}{\dim_\psi} \|W\|_F^2 \|x\|_2^2 \;.$$

From the above identity, it follows that the spectral norm of $W$ is

$$\|W\|_2 = \frac{1}{\sqrt{\dim_\psi}} \|W\|_F \tag{19}$$

For any irrep $\psi$, we can express any matrix $W$ equivariant to $\psi$ as

$$W = \sum_{k=1}^{c_\psi} w_k B_{\psi,k}$$

where $\{w_k\}_{k=1}^{c_\psi}$ are the $c_\psi$ free parameters. When $\psi$ has real type, $c_\psi = 1$ and $B_{\psi,1} = I$ is the identity matrix. When $\psi$ has complex type, $c_\psi = 2$ and $B_{\psi,1} = I_{2d}$ is the identity matrix while $B_{\psi,2} = \begin{bmatrix} 0 & -I_d \\ I_d & 0 \end{bmatrix}$, where $d = \dim_\psi /2$. Finally, if $\psi$ has quaternionic type, $c_\psi = 4$

$$B_{\psi,1} = I_{4d}, B_{\psi,2} = \begin{bmatrix} 0 & 0 & -I_d & 0 \\ 0 & 0 & 0 & -I_d \\ I_d & 0 & 0 & 0 \\ 0 & I_d & 0 & 0 \end{bmatrix}, B_{\psi,3} = \begin{bmatrix} 0 & -I_d & 0 & 0 \\ I_d & 0 & 0 & 0 \\ 0 & 0 & 0 & I_d \\ 0 & 0 & -I_d & 0 \end{bmatrix}, B_{\psi,4} = \begin{bmatrix} 0 & 0 & 0 & -I_d \\ 0 & 0 & I_d & 0 \\ 0 & -I_d & 0 & 0 \\ I_d & 0 & 0 & 0 \end{bmatrix}$$

where $d = \dim_\psi /4$. Note that, for any $\psi$ and $1 \le k \le c_\psi$, $B_{\psi,k}$ is an orthonormal matrix.

# 9 Proof of Theorem 4.1

In this section, we provide the detailed proof of Theorem 4.1. The complete version of the theorem, including all consants and dependences, is given below.

**Theorem 9.1** (Homogeneous Bounds for Equivariant Networks). *For any equivariant network, with probability $1 - \delta$, we have:*

$$\mathcal{L}(f_{\mathbb{W}}) \leq \hat{\mathcal{L}}_\gamma(f_{\mathbb{W}}) + \tag{20}$$

$$\sqrt{\frac{32e^4 B^2 \beta^2}{\gamma^2 m \eta} \left( \sum_{l=1}^{L} \sqrt{M(l,\eta)} \right)^2 \left( \sum_l \frac{\sum_{\psi,i,j} \left\| \widehat{\boldsymbol{W}}_l(\psi,i,j) \right\|_F^2 / \dim_\psi}{\|\boldsymbol{W}_l\|_2^2} \right) + \frac{\log(\xi(m)\frac{Lm^{1+1/2L}}{\delta})}{2\gamma^2 m}} \tag{21}$$

*where $\beta = (\prod_l \|\boldsymbol{W}_l\|_2)$, and*

$$M(l,\eta) := \log \left( \frac{\sum_{l=1}^{L} \sum_\psi m_{l,\psi}}{1 - \eta} \right) \max_\psi \left( 5 m_{l-1,\psi} m_{l,\psi} c_\psi \right). \tag{22}$$

## 9.1 Proof of Lemma 4.3: Perturbation Analysis

The proof of perturbation bound uses standard inequalities and given in Neyshabur et al. [2018]. We derive a bound on the spectral norm of an equivariant matrix $\boldsymbol{W}_l$.

$$
\begin{aligned}
\|\boldsymbol{W}_l \boldsymbol{x}\|_2^2 = \left\| \widehat{\boldsymbol{W}}_l \widehat{\boldsymbol{x}} \right\|_2^2 &= \sum_\psi \sum_{j=1}^{m_{l,\psi}} \left\| \widehat{\boldsymbol{W}_l \boldsymbol{x}}(\psi,j) \right\|_2^2 \\
&= \sum_\psi \sum_{j=1}^{m_{l,\psi}} \left\| \sum_{i=1}^{m_{l-1,\psi}} \widehat{\boldsymbol{W}}_l(\psi,j,i) \widehat{\boldsymbol{x}}(\psi,i) \right\|_2^2 \\
&\overset{(a)}{\leq} \sum_\psi \sum_{j=1}^{m_{l,\psi}} m_{l-1,\psi} \sum_{i=1}^{m_{l-1,\psi}} \left\| \widehat{\boldsymbol{W}}_l(\psi,j,i) \widehat{\boldsymbol{x}}(\psi,i) \right\|_2^2 \\
&\leq \sum_\psi \sum_{j=1}^{m_{l,\psi}} m_{l-1,\psi} \sum_{i=1}^{m_{l-1,\psi}} \left\| \widehat{\boldsymbol{W}}_l(\psi,j,i) \right\|_2^2 \|\widehat{\boldsymbol{x}}(\psi,i)\|_2^2 \\
&\overset{(b)}{=} \sum_\psi \sum_{j=1}^{m_{l,\psi}} m_{l-1,\psi} \sum_{i=1}^{m_{l-1,\psi}} \frac{1}{\dim_\psi} \left\| \widehat{\boldsymbol{W}}_l(\psi,j,i) \right\|_F^2 \|\widehat{\boldsymbol{x}}(\psi,i)\|_2^2 \\
&= \sum_\psi \sum_{i=1}^{m_{l-1,\psi}} \left( m_{l-1,\psi} \frac{1}{\dim_\psi} \sum_{j=1}^{m_{l,\psi}} \left\| \widehat{\boldsymbol{W}}_l(\psi,j,i) \right\|_F^2 \right) \|\widehat{\boldsymbol{x}}(\psi,i)\|_2^2 \\
&\leq \left( \max_\psi \max_{1 \leq i \leq m_{l-1,\psi}} m_{l-1,\psi} \frac{1}{\dim_\psi} \sum_{j=1}^{m_{l,\psi}} \left\| \widehat{\boldsymbol{W}}_l(\psi,j,i) \right\|_F^2 \right) \|\boldsymbol{x}\|_2^2
\end{aligned}
$$

where $(a)$ follows from Cauchy-Schwarz inequality, and $(b)$ from eq. 19, namely that for equivariant matrices, we have:

$$\|\boldsymbol{W}\|_2 = \frac{1}{\sqrt{\dim_\psi}} \|\boldsymbol{W}\|_F \tag{23}$$

Therefore, we get:

$$\|\boldsymbol{W}_l\|_2 \leq \sqrt{\max_\psi \max_{1 \leq i \leq m_{l-1,\psi}} m_{l-1,\psi} \frac{1}{\dim_\psi} \sum_{j=1}^{m_{l,\psi}} \left\| \widehat{\boldsymbol{W}}_l(\psi,j,i) \right\|_F^2} \tag{24}$$

## 9.2 Proofs of Lemma 4.4 and Tail Bound for Spectral Norms of Equivariant Kernels

*Proof.* Recall that the matrix $\widehat{\boldsymbol{U}}_l(\psi, j, i)$ has only $c_\psi$ learnable entries and that $\frac{1}{\dim_\psi} \left\| \widehat{\boldsymbol{U}}_l(\psi, j, i) \right\|_F^2$ is equal to the sum of the squares (see Supplementary 8):

$$\frac{1}{\dim_\psi} \left\| \widehat{\boldsymbol{U}}_l(\psi, j, i) \right\|_F^2 = \sum_{k=1}^{c_\psi} \left| \widehat{u_{l,i,j}}(\psi)_k \right|^2 .$$

To find tail bounds on the spectral norm, note that $\sum_{j=1}^{m_{l,\psi}} \sum_k \left| \widehat{u_{l,i,j}}(\psi)_k \right|^2$ is a $\chi^2$-random variable with $m_{l,\psi} \times c_\psi$ degrees of freedom. We will use the following inequality for bounding the tail.

**Lemma 9.2** ([Laurent and Massart, 2000, Lemma 1.]). *Let $X_1, \ldots, X_n$ be i.i.d. Gaussian random variable with zero mean and variance 1. For any vector $\boldsymbol{a} \in \mathbb{R}^n$, we have*

$$\mathbb{P}(\sum_{i=1}^n a_i(X_i^2 - 1) \geq 2 \|\boldsymbol{a}\|_2 \sqrt{x} + 2 \|\boldsymbol{a}\|_\infty x) \leq \exp(-x). \tag{25}$$

Then, using this theorem, we have:

$$\mathbb{P} \left( \frac{1}{\dim_\psi} \sum_{j=1}^{m_{l,\psi}} \left\| \widehat{\boldsymbol{U}}_l(\psi, j, i) \right\|_F^2 \geq m_{l,\psi} c_\psi \sigma^2 + 2 m_{l,\psi} c_\psi \sigma^2 \sqrt{t} + 2\sigma^2 t \right) \leq e^{-t}$$

Then:

$$\mathbb{P} \left( m_{l-1,\psi} \frac{1}{\dim_\psi} \sum_{j=1}^{m_{l,\psi}} \left\| \widehat{\boldsymbol{U}}_l(\psi, j, i) \right\|_F^2 \geq m_{l-1,\psi} \left( m_{l,\psi} c_\psi \sigma^2 + 2 m_{l,\psi} c_\psi \sigma^2 \sqrt{t} + 2\sigma^2 t \right) \right) \leq e^{-t}$$

Using the union bound:

$$\left( \sum_\psi m_{l,\psi} \right) e^{-t} \geq \mathbb{P} \left( \bigvee_\psi^{m_{l,\psi}} \bigvee_i \left[ \frac{m_{l-1,\psi}}{\dim_\psi} \sum_{j=1}^{m_{l,\psi}} \left\| \widehat{\boldsymbol{U}}_l(\psi, j, i) \right\|_F^2 \geq m_{l-1,\psi} \left( m_{l,\psi} c_\psi \sigma^2 + 2 m_{l,\psi} c_\psi \sigma^2 \sqrt{t} + 2\sigma^2 t \right) \right] \right)$$

$$\geq \mathbb{P} \left( \max_\psi \max_i^{m_{l,\psi}} \frac{m_{l-1,\psi}}{\dim_\psi} \sum_{j=1}^{m_{l,\psi}} \left\| \widehat{\boldsymbol{U}}_l(\psi, j, i) \right\|_F^2 \geq \max_\psi m_{l-1,\psi} \left( m_{l,\psi} c_\psi \sigma^2 + 2 m_{l,\psi} c_\psi \sigma^2 \sqrt{t} + 2\sigma^2 t \right) \right)$$

We can combine this bound with the bound on the spectral norm in Supplementary 9.1 to obtain a bound similar to the one in Lemma 4.4:

$$\mathbb{P} \left( \|\boldsymbol{U}_l\|_2 \geq \sigma \sqrt{\max_\psi m_{l-1,\psi} \left( m_{l,\psi} c_\psi + 2 m_{l,\psi} c_\psi \sqrt{t} + 2t \right)} \right) \leq \left( \sum_\psi m_{l,\psi} \right) e^{-t} \tag{26}$$

Using $5 m_{l-1,\psi} m_{l,\psi} c_\psi t \geq m_{l-1,\psi} (m_{l,\psi} c_\psi + 2 m_{l,\psi} c_\psi \sqrt{t} + 2t)$:

$$\mathbb{P} \left( \|\boldsymbol{U}_l\|_2 \geq \sigma \sqrt{\max_\psi \left( 5 m_{l-1,\psi} m_{l,\psi} c_\psi t \right)} \right) \leq \left( \sum_\psi m_{l,\psi} \right) e^{-t} \tag{27}$$

Using an union bound over all layers, one finds that by choosing $t = \log \left( 2 \sum_l \sum_\psi m_{l,\psi} \right)$, with probability at least $\frac{1}{2}$ the following inequality holds for every layer $l \in [L]$:

$$\|\boldsymbol{U}_l\|_2 < \sigma \sqrt{\max_\psi 5 m_{l-1,\psi} m_{l,\psi} c_\psi} \sqrt{\log \left( 2 \sum_l \sum_\psi m_{l,\psi} \right)} \tag{28}$$

$\square$

### 9.3 A KL Divergence Identity

We use the following lemma later in context of PAC-Bayes bounds.

**Lemma 9.3.** *For any two probability measures $\mu, \rho$ defined on the probability space $(\Omega, \mathcal{F})$, assume that $\mu_{|A}$ is defined as the normalized probability measure restricted to the set $A \in \mathcal{F}$. Then we have:*

$$D(\mu\|\rho) = \mu(A)D(\mu_{|A}\|\rho) + \mu(A^c)D\left(\mu_{|A^c}\|\rho\right) + h_b(\mu(A)), \tag{29}$$

*where $h_b(\cdot)$ is the binary entropy defined as $h_b(p) = -p \log p - (1-p)\log(1-p)$.*

The proof of this lemma follows from standard integral manipulations.

### 9.4 PAC-Bayesian Generalization Bound - Derandomization Technique

The PAC-Bayesian generalization lemma of 4.2 provides a bound on the generalization error of randomized classifiers drawn from the posterior distribution $Q$. To obtain a bound that holds for individual hypothesis, we need to de-randomize the bound. The following lemma, a reformulation of Neyshabur et al. [2018], provides a way of achieving this goal.

**Lemma 9.4.** *Let $\mathcal{P}$ be a finite set of priors $P$ over the hypothesis space. Let the set $\mathcal{S}_{h_0}$ be defined as:*

$$\mathcal{S}_{h_0} = \{h \in \mathcal{H} : \|h - h_0\|_\infty \leq \gamma/4\}.$$

*If the hypothesis after training is chosen using the posterior $Q$, set $\eta := Q(\mathcal{S}_{h_0})$. Then with probability $1 - \delta$, we have for all $Q$:*

$$\mathcal{L}(h_0) \leq \hat{\mathcal{L}}_\gamma(h_0) + \sqrt{\frac{\min_{P \in \mathcal{P}} D(Q\|P) + \eta \log(\xi(m)|\mathcal{P}|/\delta)}{2m\eta}}, \tag{30}$$

*where $\xi(m) := \sum_{k=0}^m \binom{m}{k}(k/m)^k(1 - k/m)^{m-k}$.*

*Proof.* PAC-Bayesian bound in Lemma 4.2 holds for a single $P$. Taking the union bound over all $P \in \mathcal{P}$, we can see that with probability at least $1 - |\mathcal{P}|\delta$ for all $P \in \mathcal{P}$ and for all $Q$ over $\mathcal{H}$, the bound in Lemma 4.2 holds. Choose the margin loss eq. 1 with margin $\gamma/2$ as the loss function. After replacing $\delta$ with $\delta/|\mathcal{P}|$, the following inequality holds with probability $1 - \delta$:

$$\mathcal{L}_{\gamma/2}(Q) \leq \hat{\mathcal{L}}_{\gamma/2}(Q) + \sqrt{\frac{\min_{P \in \mathcal{P}} D(Q\|P) + \log(\xi(m)|\mathcal{P}|/\delta)}{2m}}. \tag{31}$$

For each $Q$, define $\hat{Q}$ as:

$$\hat{Q}(h) := \begin{cases} \frac{1}{\eta}Q(h) & h \in \mathcal{S}_{h_0} \\ 0 & \text{otherwise.} \end{cases} \tag{32}$$

Since eq. 31 holds for all $Q$, we can use $\hat{Q}$ in the bound. Let $h$ be drawn from $\hat{Q}$. Then $h$ belongs to $\mathcal{S}_{h_0}$. Knowing that, we follow the argument we sketched above to obtain bound on the risks of $h_0$. Since $h \in \mathcal{S}_{h_0}$, $\|h - h_0\|_\infty \leq \gamma/4$. We have:

$$\left(h(\boldsymbol{x})[y] - \max_{j \neq y} h(\boldsymbol{x})[j]\right) \leq \left(h_0(\boldsymbol{x})[y] + \gamma/4 - \max_{j \neq y}(h_0(\boldsymbol{x})[j] - \gamma/4)\right) \tag{33}$$

$$\leq \left(h_0(\boldsymbol{x})[y] - \max_{j \neq y} h_0(\boldsymbol{x})[j]\right) + \gamma/2, \tag{34}$$

and therefore:

$$\mathcal{L}(h_0) = \mathbb{P}_{(\boldsymbol{x},y)\sim\mathcal{D}}\left(h_0(\boldsymbol{x})[y] \leq \max_{j \neq y} h_0[j]\right) \leq \mathcal{L}_{\gamma/2}(h) = \mathbb{P}_{(\boldsymbol{x},y)\sim\mathcal{D}}\left(h(\boldsymbol{x})[y] \leq \gamma/2 + \max_{j \neq y} h[j]\right). \tag{35}$$

Similarly we have:

$$\left(h_0(\boldsymbol{x})[y] - \max_{j \neq y} h_0(\boldsymbol{x})[j]\right) \leq \left(h(\boldsymbol{x})[y] - \max_{j \neq y} h(\boldsymbol{x})[j]\right) + \gamma/2, \tag{36}$$

which provides a bound on the empirical risk as follows:

$$\hat{\mathcal{L}}_{\gamma/2}(h) \le \hat{\mathcal{L}}_{\gamma}(h_0). \tag{37}$$

Since inequalities 35 and 37 hold for all $h \in \mathcal{S}_{h_0}$, we get:

$$\mathcal{L}(h_0) \le \mathcal{L}_{\gamma/2}(\hat{Q}) \quad \text{and} \quad \hat{\mathcal{L}}_{\gamma/2}(\hat{Q}) \le \hat{\mathcal{L}}_{\gamma}(h_0), \tag{38}$$

from which we immediately obtain the following de-randomized bound:

$$\mathcal{L}(h_0) \le \hat{\mathcal{L}}_{\gamma}(h_0) + \sqrt{\frac{\min_{P \in \mathcal{P}} D(\hat{Q}\|P) + \log(\xi(m)|\mathcal{P}|/\delta)}{2m}}. \tag{39}$$

Applying the lemma 9.3, we get $\eta D\left(\hat{Q}\|P\right) \le D\left(Q\|P\right)$. Using this inequality, the lemma is obtained from eq. 39.

$\square$

## 9.5   Proof of Theorem 9.1 and Generalization Bounds for Equivariant Networks

*Proof.*  We consider the perturbation bound in Eq. 7:

$$\|f_{\mathbb{W}+\mathbb{U}} - f_{\mathbb{W}}\|_\infty \le \|f_{\mathbb{W}+\mathbb{U}} - f_{\mathbb{W}}\|_2 \le eB\left(\prod_{i=1}^{L}\|\boldsymbol{W}_i\|_2\right)\sum_{i=1}^{L}\frac{\|\boldsymbol{U}_i\|_2}{\|\boldsymbol{W}_i\|_2}. \tag{40}$$

By applying the results found in Supplementary 9.2 together with a union bound argument, with probability $1 - \left(\sum_{l=1}^{L}\sum_{\psi} m_{l,\psi}\right)e^{-t}$, for all $l$, it holds (for $t > 1$):

$$\|\boldsymbol{U}_l\|_2 \le \sigma\sqrt{\max_{\psi}\left(5m_{l-1,\psi}m_{l,\psi}c_\psi t\right)}. \tag{41}$$

Therefore, with probability $1 - \left(\sum_{l=1}^{L}\sum_{\psi} m_{l,\psi}\right)e^{-t}$,

$$\|f_{\mathbb{W}+\mathbb{U}} - f_{\mathbb{W}}\|_\infty \le eB\left(\prod_{i=1}^{L}\|\boldsymbol{W}_i\|_2\right)\sum_{l=1}^{L}\frac{\sigma\sqrt{\max_{\psi}\left(5m_{l-1,\psi}m_{l,\psi}c_\psi t\right)}}{\|\boldsymbol{W}_l\|_2}. \tag{42}$$

Choosing $t = \log\left(\frac{\sum_{l=1}^{L}\sum_{\psi} m_{l,\psi}}{1-\eta}\right)$, with probability $\eta$:

$$\|f_{\mathbb{W}+\mathbb{U}} - f_{\mathbb{W}}\|_\infty \le eB\left(\prod_{i=1}^{L}\|\boldsymbol{W}_i\|_2\right)\sum_{l=1}^{L}\frac{\sigma\sqrt{\max_{\psi}\left(5m_{l-1,\psi}m_{l,\psi}c_\psi \log\left(\frac{\sum_{l=1}^{L}\sum_{\psi} m_{l,\psi}}{1-\eta}\right)\right)}}{\|\boldsymbol{W}_l\|_2}. \tag{43}$$

To reduce clutter in the next equations, we define

$$M(l,\eta) := \log\left(\frac{\sum_{l=1}^{L}\sum_{\psi} m_{l,\psi}}{1-\eta}\right)\max_{\psi}\left(5m_{l-1,\psi}m_{l,\psi}c_\psi\right) \tag{44}$$

such that the previous bound becomes:

$$\|f_{\mathbb{W}+\mathbb{U}} - f_{\mathbb{W}}\|_\infty \le eB\left(\prod_{i=1}^{L}\|\boldsymbol{W}_i\|_2\right)\sum_{l=1}^{L}\frac{\sigma\sqrt{M(l,\eta)}}{\|\boldsymbol{W}_l\|_2}. \tag{45}$$

We can then consider a set $\mathcal{P}$ of possible values for $\sigma$, such that for any $\sigma_0$ there exists a $\sigma \in \mathcal{P}$ satisfying

$$|\sigma - \sigma_0| \le \kappa\sigma_0 \tag{46}$$

with

$$\sigma_0 := \frac{\gamma}{4eB(1+\kappa)\left(\prod_{i=1}^{L}\|\boldsymbol{W}_i\|_2\right)\sum_{i=1}^{L}\frac{\sqrt{M(l,\eta)}}{\|\boldsymbol{W}_i\|_2}}. \tag{47}$$

By choosing the covariance of the distribution $Q$ from this set $\mathcal{P}$ of values for $\sigma$, we get

$$Q(\mathcal{S}_{f_{\mathbb{W}}}) \geq \eta. \tag{48}$$

Again, suppose that such a set $\mathcal{P}$ of $\sigma$ values can be chosen and that it has cardinality $|\mathcal{P}|$. We can then use the PAC-Bayesian KL-bound in Eq. 5 and choose a $\sigma$ satisfying the above inequality to obtain:

$$D(Q\|P) \leq \frac{(1+\kappa)^2 32 e^2 B^2 \left(\prod_{i=1}^{L}\|\boldsymbol{W}_i\|_2^2\right)}{(1-\kappa)^2\gamma^2}\left(\sum_{l=1}^{L}\frac{\sqrt{M(l,\eta)}}{\|\boldsymbol{W}_l\|_2}\right)^2\left(\sum_{l,\psi,i,j}\left\|\widehat{\boldsymbol{W}}_l(\psi,i,j)\right\|_F^2 / \dim_\psi\right) \tag{49}$$

$$\leq \frac{(1+\kappa)^2 32 L e^2 B^2 \left(\prod_{i=1}^{L}\|\boldsymbol{W}_i\|_2^2\right)}{(1-\kappa)^2\gamma^2}\left(\sum_{l=1}^{L}\frac{M(l,\eta)}{\|\boldsymbol{W}_l\|_2^2}\right)\left(\sum_{l,\psi,i,j}\left\|\widehat{\boldsymbol{W}}_l(\psi,i,j)\right\|_F^2 / \dim_\psi\right). \tag{50}$$

We need to construct a set $\mathcal{P}$ such that for any $\sigma_0$ there exists a $\sigma \in \mathcal{P}$ satisfying inequality 46. Using a similar argument to Neyshabur et al. [2018], it can be seen that the bound is non-vacuous if $\prod_{l=1}^{L}\|\boldsymbol{W}_l\|_2 \in [\gamma/2B, \gamma\sqrt{m}/2B]$. If the product of norms after training is outside the interval, the bound holds trivially, and any $(P,Q)$ can be chosen. Therefore, we cover only this interval. Details follow below.

In homogeneous networks, we can always rescale the weights such that the margin is not touched, and thereby change all the norms. The above generalization bound is not invariant to rescaling of weights. Here, we derive a scaling-invariant version of it. Define $\beta^L = \prod_{i=1}^{L}\|\boldsymbol{W}_i\|_2$ and normalize every layer to have norm $\beta$. Then, $\sigma_0$ can be written as:

$$\sigma_0 = \frac{\gamma}{4eB(1+\kappa)\beta^{L-1}\sum_{l=1}^{L}\sqrt{M(l,\eta)}}. \tag{51}$$

To cover the set $\sigma_0$, we require only to cover $\beta$ effectively. Again, we consider only $\beta^L > \frac{\gamma}{2B}$ and $\beta^{2L} < \left(\frac{\gamma^2 m}{4B^2}\right)$, such that the bound is not vacuous. Note that the generalization bound in Eq. 49 is simplified to (removing $\kappa$ as we cover our set differently):

$$\frac{32e^2 B^2 \beta^{2L}}{\gamma^2}\left(\sum_{l=1}^{L}\frac{\sqrt{M(l,\eta)}}{\beta}\right)^2\left(\sum_{l}\frac{\beta^2}{\|\boldsymbol{W}_l\|_2^2}\sum_{\psi,i,j}\left\|\widehat{\boldsymbol{W}}_l(\psi,i,j)\right\|_F^2 / \dim_\psi\right) \tag{52}$$

$$= \frac{32e^2 B^2 \beta^{2L}}{\gamma^2}\left(\sum_{l=1}^{L}\sqrt{M(l,\eta)}\right)^2\left(\sum_{l}\frac{\sum_{\psi,i,j}\left\|\widehat{\boldsymbol{W}}_l(\psi,i,j)\right\|_F^2 / \dim_\psi}{\|\boldsymbol{W}_l\|_2^2}\right). \tag{53}$$

Therefore, if $\beta > (\frac{\gamma\sqrt{m}}{2B})^{1/L}$ all the remaining terms are bigger than one. Specifically note that:

$$\|\boldsymbol{W}_l\|_2^2 \leq \sum_{\psi,i,j}\left\|\widehat{\boldsymbol{W}}_l(\psi,i,j)\right\|_2^2 = \sum_{\psi,i,j}\left\|\widehat{\boldsymbol{W}}_l(\psi,i,j)\right\|_F^2 / \dim_\psi \tag{54}$$

where we first used the fact that the spectral norm of a block matrix is upperbounded by the sum of the spectral norm of each block and then we used Eq. 19 to express the spectral norm of each block.

Therefore we need only to cover $\beta$ within the interval $(\frac{\gamma}{2B})^{1/L} \leq \beta \leq (\frac{\gamma\sqrt{m}}{2B})^{1/L}$ with radius $1/d(\frac{\gamma}{2B})^{1/L}$.

This choice guarantees that there is a $\tilde{\beta}$ such that $\left\|\beta - \tilde{\beta}\right\| \leq \beta/d$. From which, it can be concluded that $1/e\beta^{L-1} \leq \hat{\beta}^{L-1} \leq e\beta^{L-1}$. Using a cover of size $Lm^{1/2L}$ provides the desired cover. $\qquad\square$

# 10 Derivations for Group Convolutional Networks

In this section, we provide a special case of Theorem 9.1 for group convolutional networks.

In a group convolution architecture, we assume the features at the $l$-th layer are $C_l = c_l|H|$-dimensional vectors representing a $c_l$-channels signal over $H$. This means $\rho_l$ contains $c_l$ copies of the regular representation. The linear layer consists of a matrix $\boldsymbol{W}_l$ with a block structure as in Eq. 18, where the block $\boldsymbol{W}_{l,(i,j)}$ is an $H$-circulant matrix mapping the $i$-th input channel to the $j$-th output channel via group convolution with a filter $\boldsymbol{w}_{l,i,j} \in \mathbb{R}^{|H|}$. The parametrization of this block is in term of the Fourier transform of $\boldsymbol{w}_{l,i,j}$, i.e. the coefficient in the matrices $\widehat{\boldsymbol{w}_{l,i,j}}(\psi)$, indexed by the irreps $\psi \in \hat{H}$. Since we use real irreps, we only need the first $\dim_\psi /c_\psi$ columns of $\widehat{\boldsymbol{w}_{l,i,j}}(\psi)$, so we assume it is a $\dim_\psi \times \frac{\dim_\psi}{c_\psi}$ matrix. Similarly, the Fourier transform of a single channel $\boldsymbol{x}_{l,i} \in \mathbb{R}^{|H|}$ only contains the first $\dim_\psi /c_\psi$ columns of $\widehat{x}_{l,i}(\psi)$. In the notation used before, this means that $\psi$ is contained in each channel $\dim_\psi /c_\psi$ times and, therefore, for each (intermediate) layer $l$ and irrep $\psi$, it holds that:

$$m_{l,\psi} = c_l \dim_\psi /c_\psi.$$

Note also that, for each block $(i,j)$, the matrix $\widehat{\boldsymbol{w}_{l,i,j}}(\psi) \in \mathbb{R}^{\dim_\psi \times \frac{\dim_\psi}{c_\psi}}$ contains the $c_\psi$ coefficients in $\widehat{\boldsymbol{w}_{l,\cdot,\cdot}}(\psi)$ associated to each of the $\frac{\dim_\psi}{c_\psi} \times \frac{\dim_\psi}{c_\psi}$ pairs of input and output occurrences of $\psi$ in $\boldsymbol{x}_{l-1,i}$ and $\boldsymbol{x}_{l,j}$.

Assuming a group-convolutional architecture, we can use the identity $m_{l,\psi} = c_l \frac{\dim_\psi}{c_\psi}$. Define:

$$D_H := \max_\psi \frac{\dim_\psi^2}{c_\psi} \tag{55}$$

$$E_H := \sum_\psi \frac{\dim_\psi}{c_\psi}. \tag{56}$$

In case of a group convolution architecture, we get:

$$M(l,\eta) = \log\left(\frac{E_H \sum_{l=1}^L c_l}{1-\eta}\right) 5c_l c_{l-1} D_H \tag{57}$$

and the bound in Theorem 9.1 becomes:

$$\mathcal{L}_{\mathcal{D}}(f_{\mathbb{W}}) \le \hat{\mathcal{L}}_\gamma(f_{\mathbb{W}})+$$

$$\sqrt{\frac{32e^4 B^2 \beta^2}{\gamma^2 m \eta}\left(\sum_{l=1}^L \sqrt{5c_l c_{l-1}}\right)^2 D_H \log\left(\frac{E_H \sum_{l=1}^L c_l}{1-\eta}\right)\left(\sum_l \frac{\sum_{\psi,i,j}\left\|\widehat{\boldsymbol{w}_{l,j,i}}(\psi)\right\|_F^2}{\|\boldsymbol{W}_l\|_2^2}\right) + \frac{\log(\xi(m)^{\frac{Lm^{1+1/2L}}{\delta}})}{2\gamma^2 m}}$$

By choosing $\eta = 1/2$ and by defining $Q_H = \left(\sum_{l=1}^L \sqrt{c_{l-1}c_l}\right)^2 D_H \log 2E_H \sum_{l=1}^L c_{l-1}$, this bound corresponds to Theorem 10.1.

**Theorem 10.1** (Generalization Bounds for Group Convolutional Networks). *For any group convolutional network, with high probability, we have:*

$$\mathcal{L}_{\mathcal{D}}(f_{\mathbb{W}}) \le \hat{\mathcal{L}}_\gamma(f_{\mathbb{W}}) + \tilde{\mathcal{O}}\left(\sqrt{\frac{\left(\prod_{l=1}^L \|\boldsymbol{W}_l\|_2^2\right) Q_H \left(\sum_l^L \frac{\sum_{\psi,i,j}\left\|\widehat{\boldsymbol{w}_{l,i,j}}(\psi)\right\|_F^2}{\|\boldsymbol{W}_l\|_2^2}\right)}{\gamma^2 m}}\right)$$

*with $Q_H = \left(\sum_{l=1}^L \sqrt{c_{l-1}c_l}\right)^2 D_H \log 2E_H \sum_{l=1}^L c_{l-1}$.*

First, note that the constant factor $D_H$ is at most 2 for commutative groups (all subgroups of $\mathrm{SO}(2)$) and at most 4 for other subgroups of $\mathrm{O}(2)$. In practical networks, the number channels are inversely scaled with the group size. This means that the $c_l|H|$ are kept constant for different values of $|H|$.

In terms of generalization error, this implies an improvement of order $1/|H|$ not withstanding the impact of group size on the ratio of Frobenius and Spectral norms.

It is worth mentioning an intermediate result for proving the above theorem. We can adapt the tail bound of eq. 11 to perturbations for group convolutional networks:

$$\|\boldsymbol{U}_l\|_2 < \sigma \sqrt{5c_{l-1}c_l \left( \max_\psi \frac{\dim_\psi^2}{c_\psi} \right) \log \left( 2 \sum_l c_1 \sum_\psi \frac{\dim_\psi}{c_\psi} \right)}.$$

which is equal to Eq. 11 when defining $D_H = \max_\psi \frac{\dim_\psi^2}{c_\psi}$ and $E_H = \sum_\psi \frac{\dim_\psi}{c_\psi}$. The bound can be further simplified by noting that $|H| = \sum_\psi \dim_\psi^2 /c_\psi \geq \sum_\psi \dim_\psi /c_\psi = E_H$:

$$\|\boldsymbol{U}_l\|_2 < \sigma \sqrt{5c_{l-1}c_l D_H \log \left( 2|H| \sum_l c_l \right)}.$$

This tail bound can be of independent interest.

## 11   Comparison to Neyshabur et al. [2018]

In this section, we try to directly adapt the proof of Neyshabur et al. [2018] to show the benefit of working in space of irreps.

First, note that the Forbenius norm of kernels is an upper-bound on the norm of parameters. To see this, similar to Neyshabur et al. [2018], define $h = \max_l C_l$, where $C_l = c_l|H|$ is the effective number of channels, as the upper bound on the number of channels, and similarly $h_{|H|} = \max_l c_l = h/|H|$. In the hidden layers and the input layer,

$$\sum_\psi \left\| \widehat{w_{l,i,j}}(\psi) \right\|_F^2 \leq \sum_\psi \dim_\psi \left\| \widehat{w_{l,i,j}}(\psi) \right\|_F^2 = |H| \left\| \boldsymbol{w}_{l,i,j} \right\|_2^2 = \left\| \boldsymbol{W}_{l,(j,i)} \right\|_F^2$$

and, therefore, $\sum_{i,j,\psi} \left\| \widehat{w_{l,i,j}}(\psi) \right\|_F^2 \leq \|\boldsymbol{W}_l\|_F^2$. By also upperbounding $c_l = C_l/|H| \leq h/|H|$ and ignoring the constant factors, the result in Theorem 10.1 becomes

$$\mathcal{L}_\mathcal{D}(f_\mathbb{W}) \leq \hat{\mathcal{L}}_\gamma(f_\mathbb{W}) + \tilde{\mathcal{O}} \left( \frac{1}{|H|} \sqrt{\frac{\left( \prod_{l=1}^L \|\boldsymbol{W}_l\|_2^2 \right) L^2 h^2 D_H \log(2Lh\frac{E_H}{|H|}) \left( \sum_l^L \frac{\|\boldsymbol{W}_l\|_F^2}{\|\boldsymbol{W}_l\|_2^2} \right)}{\gamma^2 m}} \right)$$

which can be further simplified to

$$\mathcal{L}_\mathcal{D}(f_\mathbb{W}) \leq \hat{\mathcal{L}}_\gamma(f_\mathbb{W}) + \tilde{\mathcal{O}} \left( \frac{1}{|H|} \sqrt{\frac{\left( \prod_{l=1}^L \|\boldsymbol{W}_l\|_2^2 \right) L^2 h^2 \log(2Lh) \left( \sum_l^L \frac{\|\boldsymbol{W}_l\|_F^2}{\|\boldsymbol{W}_l\|_2^2} \right)}{\gamma^2 m}} \right)$$

by assuming $D_H$ is approximately constant and by noting $E_H \leq |H|$.

This bound looks very similar to the one proposed in Neyshabur et al. [2018], with the difference that it contains 1) $h^2$ instead of $h$ and 2) an additional term $\frac{1}{|H|}$. The additional $h$ factor is constant in our experiments as we always consider fixed architectures when changing the equivariance group $H$, so we ignore it in this discussion. Instead, we want to drive the attention to the $\frac{1}{|H|}$ term.

The presence of this $\frac{1}{|H|}$ in our bound implies that, when changing the equivariance group $H$ for a fixed architecture, the bound from Neyshabur et al. [2018] scales at least $|H|$ times worse than ours. In the experiments, we empirically observe that our bound is approximately scaling like $\frac{1}{\sqrt{|H|}}$. This can be explained by the fact that with larger group size, the ratio of Frobenious norm and spectral norm increases similar to $O(|H|)$, thereby increasing the generalization error by $O(\sqrt{|H|})$. This

leads to an overall scaling of $O(1/\sqrt{|H|})$ observed by our experiments too. On the other hand, it follows that the bound from Neyshabur et al. [2018] scales like $\sqrt{H}$, which is undesirable.

However, it is still possible to adapt the method in Neyshabur et al. [2018] by considering, as done in this work, perturbations only in the equivariant subspace of the weights. The main difference with respect to our method is how the spectral norm of the perturbation matrices $\|U_l\|_2$ is bounded. In the rest of this section, we follow this strategy to derive an alternative generalization bound.

The strategy used in Neyshabur et al. [2018] can be adapted to any parametrization of the linear layers. In particular, assume $c_{l-1}$ and $c_l$ input and output channels and $W_l = \sum_k w_{l,k} B_{l,k}$ for some set of matrices $\{B_{l,k} \in \mathbb{R}^{c_l \times c_{l-1}}\}$ forming a basis for the linear layer $W_l$.

**Theorem 11.1** (Tropp [2012]). *Assume $\{u_{l,k}\}_k$ are i.i.d. random variables with a standard gaussian distribution $\mathcal{N}(0, 1)$. Define the variance parameter*

$$v_l^2 = \max \left\{ \left\| \sum_k B_{l,k}^* B_{l,k} \right\|_2 , \left\| \sum_k B_{l,k} B_{l,k}^* \right\|_2 \right\} .$$

*Then, for any $t \geq 0$:*

$$\mathbb{P}\left( \left\| \sum_k u_{l,k} B_{l,k} \right\|_2 \geq t \right) \leq (c_l + c_{l-1}) e^{-\frac{t^2}{2v^2}}$$

If perturbation is done on the weights $\{w_{l,k}\}_k$ with i.i.d. $\{u_{l,k} \sim \mathcal{N}(0, \sigma^2)\}_k$ and $h = \max c_l, c_{l-1}$, this result can be used to bound the spectral norm of the perturbation matrix as

$$\|U_l\|_2 \leq \sigma \sqrt{2 v_l^2 \sigma^2 \log(4h)}$$

Using an union bound over all layers $l \in [L]$, one gets:

$$\|U_l\|_2 \leq \sqrt{2 v^2 \sigma^2 \log(4hL)}$$

where $v^2 = \max_l v_l^2$.

In particular, in Neyshabur et al. [2018], the parametrization $W_l = \sum_{i,j} w_{l,i,j} E_{i,j}$ was considered, where $w_{l,i,j}$ is the entry of $W_l$ at row $j$ and column $i$ while $E_{i,j}$ is a matrix of the same shape containing 0 everywhere but 1 at position $(i, j)$. In that case, $v^2 = h$.

We can express the basis considered in this work as

$$W_l = \sum_{\psi,i,j,k} \widehat{w_{l,i,j}}(\psi)_k B_{l,\psi,i,j,k} ,$$

where $\widehat{w_{l,i,j}}(\psi)_k$ is the $k$-th entry of the vector $\widehat{w_{l,i,j}}(\psi)$ containing the $c_\psi$ learnable coefficients which parametrize the block $\widehat{W_l}(\psi, i, j)$. Indeed, recall that a block $\widehat{W_l}(\psi, i, j)$ always has one of the three forms described in Supplementary 8 and can be written as $\widehat{W_l}(\psi, i, j) = \sum_{k=1}^{c_\psi} \widehat{w_l}(\psi, i, j)_k B_{\psi,k}$. The matrix $B_{l,\psi,i,j,k}$ is simply a zero matrix containing the matrix $B_{\psi,k}$ at the same position of the corresponding block $\widehat{W_l}(\psi, i, j)$. By using the structure of the matrices $B_{l,\psi,i,j,k}$ and the orthonormality of all matrices $B_{\psi,k}$, one can show that in this case $v_l^2 = \max_\psi c_\psi \max_l m_{l,\psi}$. When using a group convolution architecture, because $m_{l,\psi} = c_l \frac{\dim_\psi}{c_\psi}$, this becomes $v_l^2 = \max_\psi \dim_\psi h_{|H|} = \max_\psi \dim_\psi h/|H|$. We can use this result in a similar manner to obtain a bound on the generalization error: Assuming again that $\max_\psi \dim_\psi$ is a constant factor, it follows that the bound on the spectral norm in our parametrization is $|H|$ times tighter than the one obtained in Neyshabur et al. [2018].

$$\mathcal{L}_\mathcal{D}(f_\mathbb{W}) \leq \hat{\mathcal{L}}_\gamma(f_\mathbb{W}) + \tilde{\mathcal{O}}\left( \frac{1}{\sqrt{|H|}} \sqrt{\frac{(\max_\psi \dim_\psi) L^2 h \log(2Lh) \left(\prod_{l=1}^L \|W_l\|_2^2\right) \left(\sum_l \frac{\|W_l\|_F^2}{\|W_l\|_2^2}\right)}{\gamma^2 m}} \right)$$

(58)

This bound explicitly shows the scaling of the generalization error as $\frac{1}{\sqrt{|H|}}$ which we also verified experimentally. A similar scaling was reported in Sokolić et al. [2017a], however it was mainly based on the assumption that the transformations change the input space in a considerable way. However, in our experiments we observe the bound in Theorem 9.1 empirically scales like $\frac{1}{\sqrt{|H|}}$. Because the bound in Eq. 58 is at least $\frac{1}{\sqrt{|H|}}$ times worst, we do not expect it to correlate significantly with the equivariance group $H$. We also verify this hypothesis empirically in our experiments. See Fig. 2 and 3 for a comparison of the two bounds on different continuous synthetic datasets.

A similar strategy could also be applied by considering bases for group convolution layers directly in terms of the filters parametrizing each $H$-circulant matrix (i.e. by performing the perturbation analysis in the group domain). More precisely, one could consider a basis

$$\boldsymbol{W}_l = \sum_{i,j,k} \boldsymbol{w}_{l,i,j,k} \boldsymbol{C}_{l,i,j,k} \ ,$$

where $\boldsymbol{w}_{l,i,j} \in \mathbb{R}^{|H|}$ and $\boldsymbol{w}_{l,i,j,k}$ is its $k$-th entry. $\boldsymbol{C}_{l,i,j,k}$ is then a zero matrix containing the a base circulant matrix $C_{H,k}$ in the block $(i,j)$. The matrix $C_{H,k}$ is an $H$-circulant matrix generated by the vector $\boldsymbol{e}_k = (0,\ldots,0,1,0,\ldots 0)^T \in \mathbb{R}^{|H|}$ (with 1 in the $k$-th entry). For instance, for $H = \mathrm{C}_3$

$$\{C_{\mathrm{C}_3,k}\}_{k=1}^3 = \left\{ \begin{bmatrix} 1 & 0 & 0 \\ 0 & 1 & 0 \\ 0 & 0 & 1 \end{bmatrix}, \begin{bmatrix} 0 & 1 & 0 \\ 0 & 0 & 1 \\ 1 & 0 & 0 \end{bmatrix}, \begin{bmatrix} 0 & 0 & 1 \\ 1 & 0 & 0 \\ 0 & 1 & 0 \end{bmatrix} \right\} \ .$$

In this parametrization, $v_l^2 = |H| \max\{\mathrm{c}_l, \mathrm{c}_{l-1}\}$ and, therefore, $v^2 = h$ as in Neyshabur et al. [2018]. Indeed, this leads exactly to their same bound, which we have already discussed at the beginning of this section.

## 12   Experiments

We experiment on synthetic datasets characterized by discrete or continuous symmetries as well as with transformed MNIST 12K and CIFAR10 datasets.

We choose a *discrete* subgroup $H < G$ and build a shallow MLP classifier equivariant to $H$. In particular, denoting as $\rho_0$ the representation of $G$ acting on the samples in a $G$ symmetric dataset, the input representation of the MLP is chosen to be $\mathrm{Res}_H^G \rho_0$ while in the hidden layers we use multiple copies of the regular representation $\rho_{\mathrm{reg}}^H$ of $H$, making the linear layers effectively group convolutions over $H$. For different choices of $H$, we preserve the size of each layer, which means that the number of parameters of the model is proportional to $\frac{1}{|H|}$. Note that a larger group $H$ results in a higher level of symmetry for the network but also in a loss in capacity, as less channels are allocated per each group element.

We train each model on a fixed training set until the model correctly classifies $99\%$ of it with a margin greater than $\gamma$. We use $\gamma = 10$ in the synthetic datasets and $\gamma = 2$ on the image datasets. In the synthetic datasets, we test the models on a testset consisting of $10K$ fixed samples from the full distribution $\mathbb{P}_G$ in order to measure the generalization on the whole symmetry group $G$.

The MLP used in the experiments on the synthetic datasets consists of 3 linear layers, alternated with ReLU non-linearities, and has 2048 and 512 channels in the intermediate features. In the MNIST experiments, we use 4 linear layers with 1024, 512 and 512 channels in the intermediate features. Finally, in the CIFAR experiments we use 4 layers with 2048, 1536 and 512 channels. The first linear layer of the last two models uses steerable filters from Weiler and Cesa [2019] to process the input images. The models are trained with *Adam* without any weight decay or additional regularization. Neither batch normalization nor dropout are used in the models.

To handle input images, we use an $E(2)$-convolution layer as described in Weiler and Cesa [2019]: with a filter as large as the input image, the output produced is a single vector which is only $H$ equivariant. The implementation of the kernel constraint from Weiler and Cesa [2019] automatically performs the necessary band-limiting and implicitly decomposes the image as a signal over the group $H$ considered.

## 12.1 Synthetic Datasets

**Continuous Symmetries**  For $G = \mathrm{SO}(2)$, the data distribution is defined over a $D$-dimensional torus $\mathcal{T}^D$ embedded in a $2D$ dimensional Euclidean space. The action of an element $r_\theta \in \mathrm{SO}(2)$ simultaneously rotates the $D$ circles in $\mathcal{T}^D$, potentially at different integer frequencies in each of them. Note that each circle is isomorphic to $\mathrm{SO}(2)$. We generate one point in the first circle and two points in all the others. Then, we generate $2^{D-1}$ representative points $\boldsymbol{X}$ by tacking every combination of points from each circle. We assign a random label in $\{-1, 1\}$ to each of them. The quotient distribution $\mathcal{S}_0$ is defined by sampling a random representative in $\boldsymbol{X}$ and adding some noise. In practice, we add Gaussian noise on each of the $D$ circles and, then, project each of them on the unit circle. This is the training augmentation $\mathbb{P}_S$. We finally add additional small Gaussian noise. Fig. 7 shows a projection of the dataset for $D = 2$. In our experiments we use $D = 6$. We also build

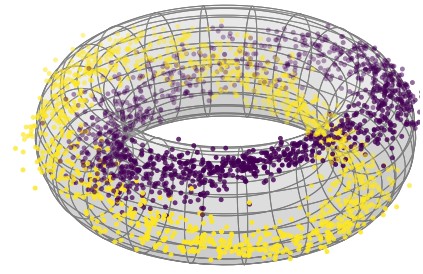
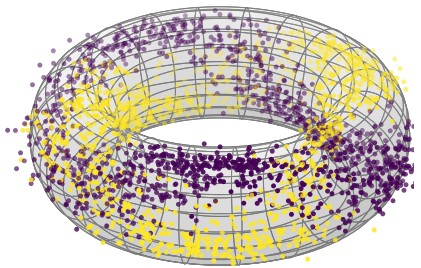

(a) $\mathrm{SO}(2)$ rotates both circles with frequency 1

(b) $\mathrm{SO}(2)$ rotates the largest circle (horizontal) with frequency 1 and the smallest (vertical) with frequency 2.

Figure 7: $3D$ projection of $\mathrm{SO}(2)$ synthetic datasets for $D = 2$.

a similar dataset for $\mathrm{O}(2)$. Here, each circle is replaced by a pair of circles such that the action of the reflection moves the points from one to the other. Each pair of circles is isomorphic to the group $\mathrm{O}(2)$ itself. We fix two points per pair such that we still have $2^{D-1}$ representative points but embedded in a $4D$-dimensional space.

**Discrete Symmetries**  We consider a group $G$ of $N$ discrete rotations and, possibly, reflections. We build a dataset as described for $\mathrm{O}(2)$ using $D = N$ pairs, each associated with a different frequency from 1 to $F = \lfloor N/2 \rfloor$. We associate all the $2^{F-1}$ representative points in $\boldsymbol{X}$ with the label $+1$. If we do not require symmetry to reflections (i.e., $G = \mathrm{C}_N$), we generate $2^{F-1}$ new representative points by rotating those in $\boldsymbol{X}$ by $\pi/N$ and associating them with the label $-1$. If symmetry to reflections is necessary ($G = \mathrm{D}_N$), we generate $3 \cdot 2^{F-1}$ new representative points by i) rotating $\boldsymbol{X}$ by $\pi/N$, ii) by mirroring it or iii) by doing both. In the first two cases, we associate the points with the label $-1$, in the last with $+1$. Note that a model equivariant to $D_{2N}$ or $C_{2N}$ will be invariant to (all or part of) the transformations we used to generate the labels and, therefore, will not be able to distinguish the two classes.

## 12.2  Dependency on the group size $|H|$

In this section we study the effect of the size $|H|$ of the equivariance group on the generalization error. In particular, we hypothesize that the generalization errors is proportional to the quantity $\frac{1}{\sqrt{|H|}}$ as previously observed in Sokolić et al. [2017a], so we investigate the correlation between these two terms in different datasets.

We consider the synthetic datasets with continuous and discrete symmetries (for different frequencies $F$ or rotation orders $M$) in Fig. 8 and the images datasets (MNIST and CIFAR10) in Fig. 9. In

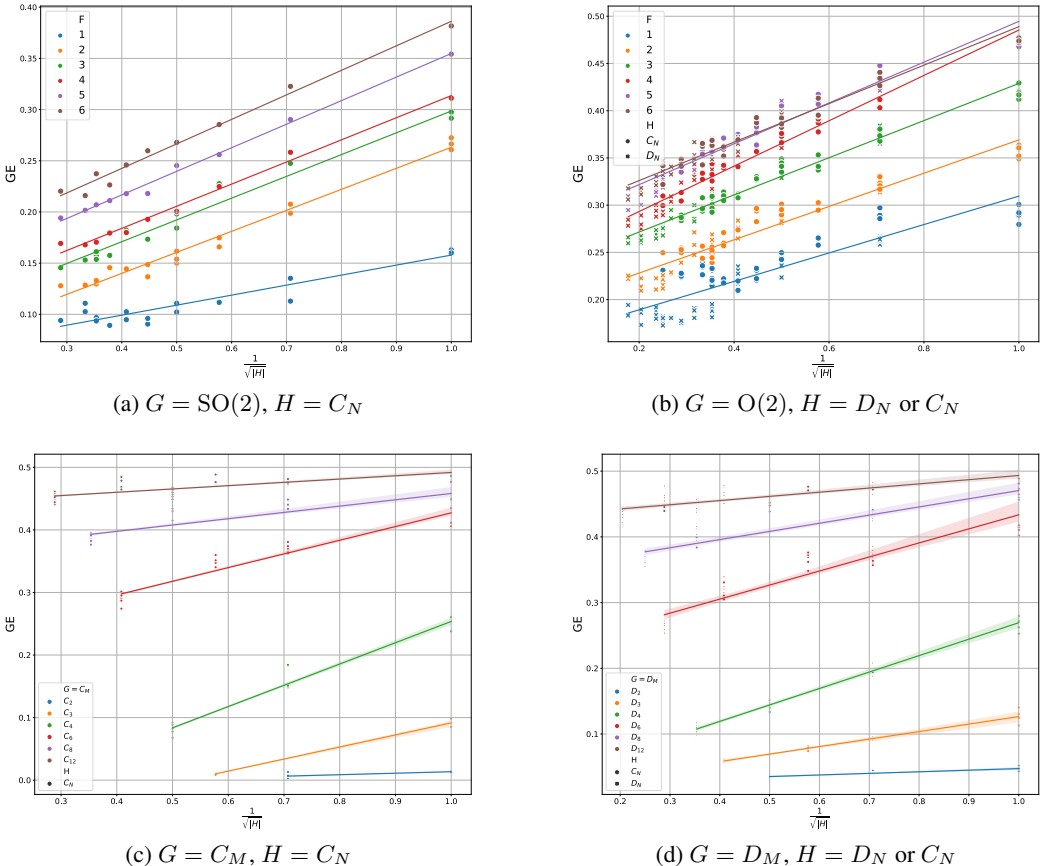

Figure 8: Empirical generalization error (GE) vs $\frac{1}{\sqrt{|H|}}$ for different synthetic datasets, with continuous or discrete symmetries $G$. In the continuous symmetries case (first row), $F$ indicates the maximum rotational frequency in the data and, therefore, each color ($F$) corresponds to a different dataset. In the discrete symmetries case (second row), $M$ indicates the the number of rotations in the discrete symmetry. The visualizations shows a strong correlation between the two terms. The correlation is weaker on low frequency $\mathrm{O}(2)$ datasets because of the different behaviours of $\mathrm{D}_N$ and $C_{2N}$.

all cases, we observe a strong correlation. However, we note that the slope of the lines varies over different versions of the same dataset or when changing the training set size $m$. In particular, in Fig. 8(first line), the slope grows when increasing the frequency $F$ in the data. In the discrete symmetries case, Fig. 8(second line), the slope increases with the symmetry group's size $|G|$ but then decreases at the largest values of $|G|$. In the $\mathrm{O}(2)$ synthetic data (Fig. 8(b)), the linear correlation is weaker for low values of $F$ as the generalization does not depend only on the size $|H|$ of the group anymore. Indeed, the groups $C_{2N}$ and $\mathrm{D}_N$ have both size $|H| = 2N$. However, when the data only contains low frequency features, considering 2 times more rotations ($C_{2N}$) becomes eventually unnecessary while introducing reflection equivariance ($\mathrm{D}_N$) allows the model to generalize over a whole new set of transformations. Overall, the

## 12.3  PAC-Bayesian Bound

In this section, we focus on the study of the de-randomized PAC-Bayes bound from Sec. 4 in different contexts. As reported in Nagarajan and Kolter [2019], bounds based on the spectral norm of the weight matrices tend to grow with the dataset size $m$ in practice. Our perturbation bound falls in this same category and, therefore, we observe a similar behaviour. Moreover, such bounds are generally vacuous, i.e. greater than 1. For these reasons, we do not expect the bound derived to accurately predict the generalization error or describe the effect of different training sizes. Instead, we are

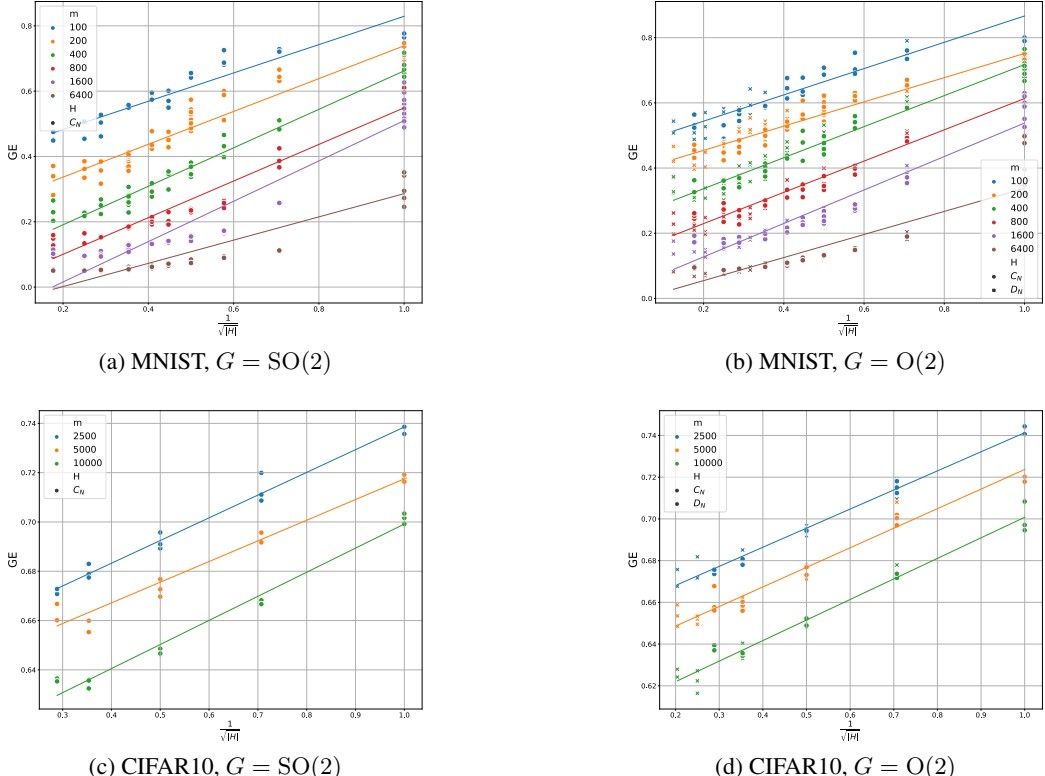

Figure 9: Empirical generalization error (GE) vs $\frac{1}{\sqrt{|H|}}$ on transformed MNIST and CIFAR10 datasets. All settings show a strong correlation between the two terms.

interested in how well the bound correlates with the generalization error and explains the effect of equivariance. Therefore, we study the correlation between the estimated bound and the generalization error in different settings and datasets when different equivariance groups $H$ are used.

In Fig. 10 we look at the correlation between our bound and the generalization error on $O(2)$-MNIST and on the $SO(2)$ symmetric synthetic dataset for different training set sizes. Nevertheless, we observe a good correlation between them when considering a fixed training set size $m$, i.e. by looking at each colored sequence independently. This is the relation we are most interested in and which we now explore further.

In Fig. 2(first row) and 3(first row) we have already observed this correlation on 6 versions of the $O(2)$ and the $SO(2)$ symmetric synthetic datasets. In Fig. 11, we repeat a similar study on the other three types of datasets.

We notice that the results on the synthetic datasets (both with continuous symmetries as in Fig. 2 and 3 and with discrete symmetries as in the first row of Fig. 11) show a better linear correlation with the generalization error. Conversely, the results on the image datasets are in log scale on the $Y$ axis, suggesting a superlinear scaling of the bound with the generalization error. In light of the observations in Supplementary 12.2, this also implies that the bound does not scale as $\frac{1}{\sqrt{|H|}}$ on these two datasets.

It also follows that the alternative bounds considered in Supplementary 11 are not worse here. However, in the synthetic dataset with discrete symmetries, the same observation from Section 5 apply. Indeed, in Fig. 12, we repeat the experiments in Fig. 11(b) but use the alternative bound from Supplementary 11. As already shown in Fig. 2(second row) and 3(second row) for the continuous symmetry synthetic datasets, this bound does not capture the effect of different equivariance groups $H$.

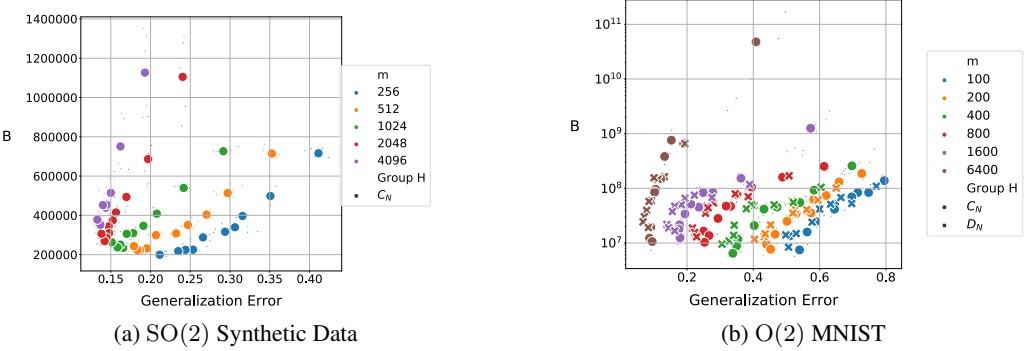

(a) SO(2) Synthetic Data  (b) O(2) MNIST

Figure 10: Bound ($B$) from Theorem 9.1 vs Generalization Error (GE) of different $H$-equivariant models ($H = C_N$ or $H = D_N$) on O(2)-MNIST and on the SO(2) synthetic dataset. Different colors represent different training set sizes $m$. As expected, the bound does not explain the effect of the training set size and as it grows with $m$.

### 12.4 PAC-Bayes Bound During Training

In this section, we study how the bound in Section 4.2 evolves during training. In particular, we consider a few different equivariance groups $H$, and we train them on the original labels as well as on random labels.

Fig. 13 reports the results of these experiments on the two of the continuous synthetic datasets. We observe that the bounds grow during training for all models. This happens using both the original and the random labels. Moreover, the bound computed with the randomly labeled training sets grows larger than the one computed on models trained on the original labels.

### 12.5 PAC-Bayes Bound on Randomly Labelled Data

Zhang et al. [2017] has shown that neural networks can fit randomly labelled training sets, obtaining arbitrarily bad generalization error. We expect a useful generalization bound to be consistent with this result and, therefore, be larger when computed on models trained on random labels. We have already observed in Supplementary 12.4 that the bound tends to grow faster while training on random datasets. Here, we perform a more extensive comparison on the two variations of the MNIST dataset. In Fig. 14, we compare the bound of different models trained on the original labels (circles) or on random ones (crosses). Training is performed on an augmented dataset, which means that each image is also transformed with a random element $g \in G$. As expected all models trained on the random labels have generalization error equals to $0.9$ (like a random classifier). Fixing a model (same $H$, i.e. same color), we observe that the bound computed over models trained on random labels is always higher. On larger training sets, we obtain higher train errors; we think this is due to the limited size, and therefore, capacity of the model used.

We perform a similar experiment on the synthetic $G = \mathrm{SO}(2)$ dataset in Fig. 15. Again, we are able to fit a randomly labeled training set with equivariant models, given sufficiently large models. Since this is a binary classification task, the models trained on the random labels have $GE = 0.5$ generalization error.

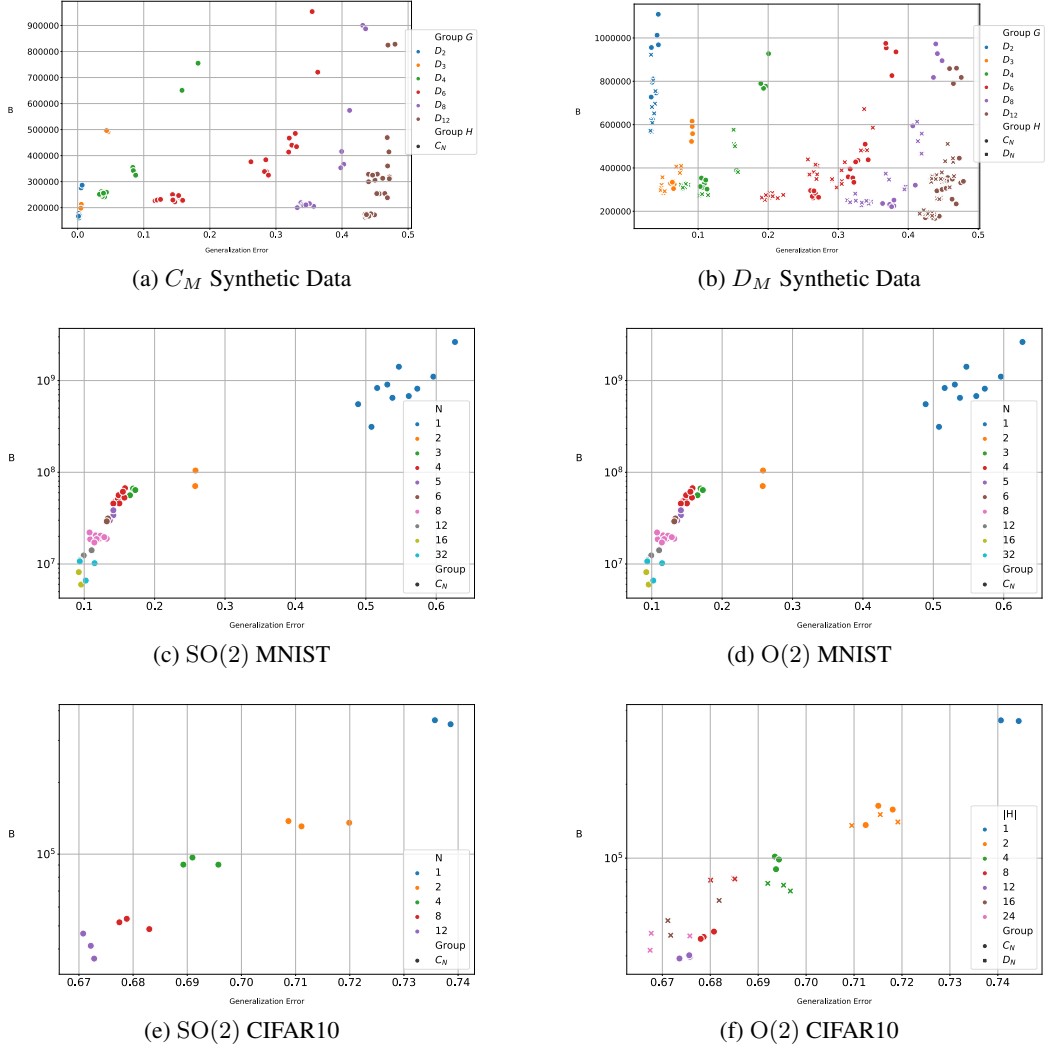

(a) $C_M$ Synthetic Data

(b) $D_M$ Synthetic Data

(c) SO(2) MNIST

(d) O(2) MNIST

(e) SO(2) CIFAR10

(f) O(2) CIFAR10

Figure 11: Bound ($B$) from Theorem 9.1 vs Generalization Error (GE) of different $H$-equivariant models on different datasets. We always use $\mathbb{P}_S = \mathbb{P}_G$. In the synthetic ones, different colors represent different discrete symmetry groups $G$, where $M$ is the number of rotations in $G = \mathrm{D}_M$ or $G = \mathrm{C}_M$; higher values of $M$ define more complex datasets.

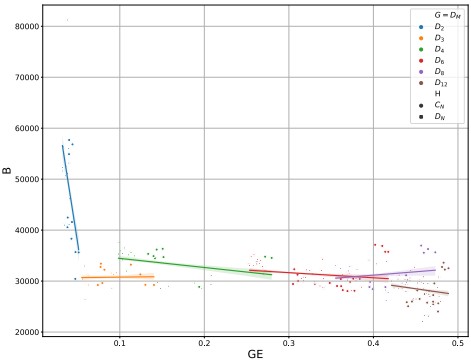

Figure 12: Alternative bound ($B$) from Supplementary 11 vs Generalization Error (GE) of different $H$-equivariant models on synthetic dataset with discrete $\mathrm{D}_M$ symmetries. As in the continuous symmetry cases, this bound does not capture the effect of $H$ equivariance.

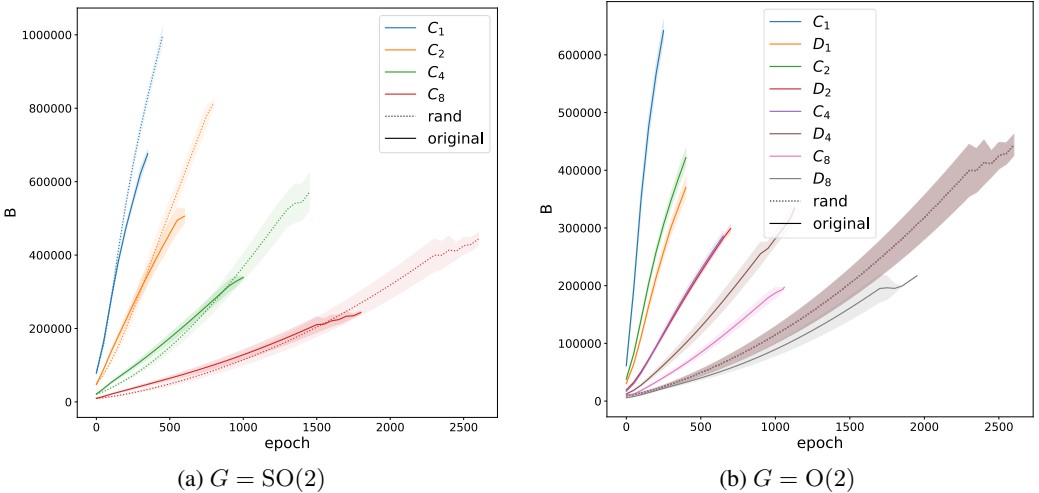

(a) $G = \mathrm{SO}(2)$        (b) $G = \mathrm{O}(2)$

Figure 13: Evolution of the PAC-Bayes Bound ($B$) during the training of $H$-equivariant models on the $\mathrm{SO}(2)$ and $\mathrm{O}(2)$ datasets with $F = 3$ and $m = 1024$ examples. Dashed lines represent the models trained on random labels.

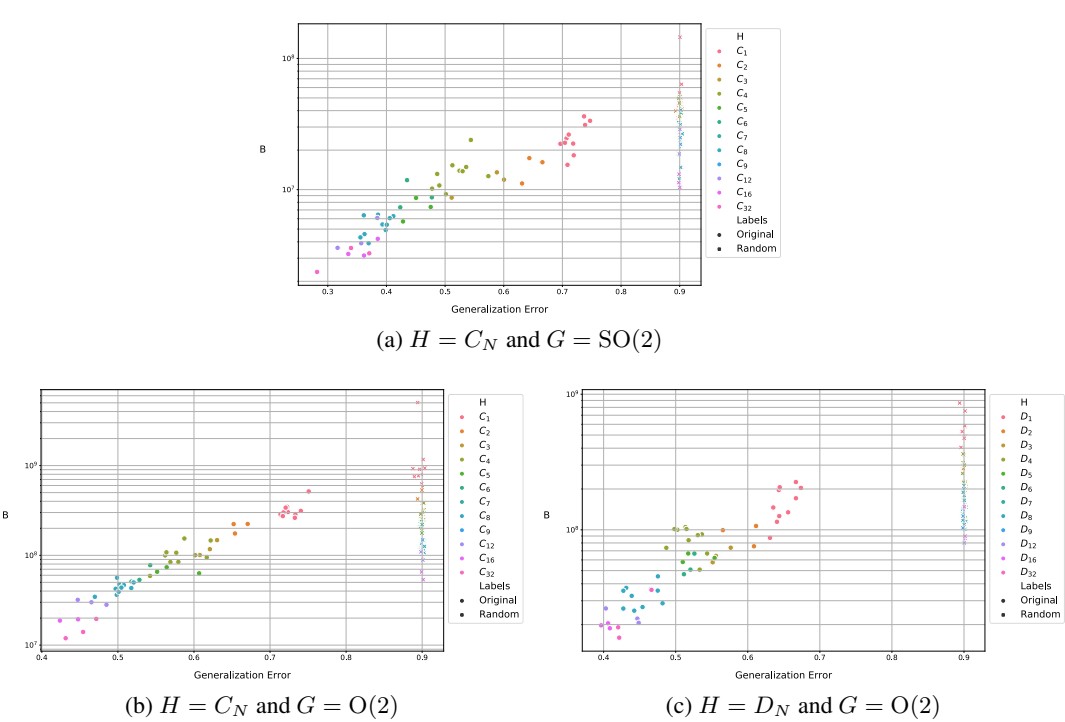

(a) $H = C_N$ and $G = \mathrm{SO}(2)$

(b) $H = C_N$ and $G = \mathrm{O}(2)$        (c) $H = D_N$ and $G = \mathrm{O}(2)$

Figure 14: PAC-Bayes Bound ($B$) from Theorem 9.1 versus Generalization Error (GE) of different $H$-equivariant models on different $G$-MNIST datasets ($G = \mathrm{SO}(2)$ or $G = \mathrm{O}(2)$) when the models are trained with the real labels or with random labels. Crosses represent models trained on random labels, while circles represent models trained on the original labels. Note that for the same training setting, the bound computed on the models trained on random labels is always higher.

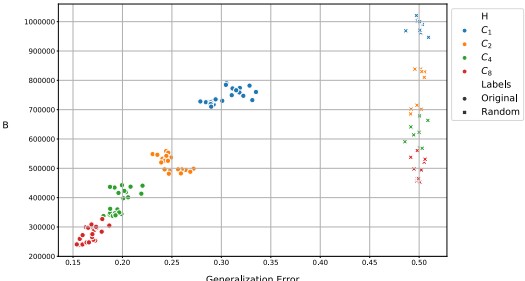

Figure 15: PAC-Bayes Bound ($B$) versus Generalization Error (GE) of different $\mathrm{C}_N$-equivariant models on the synthetic continuous $G = \mathrm{SO}(2)$ datasets when trained with the real labels or with random labels. Crosses represent models trained on random labels, while circles represent models trained on the original labels. We used frequency $F = 3$ and $F = 4$ and $m = 1024$.