# OpenReview forum: "A PAC-Bayesian Generalization Bound for Equivariant Networks"
_NeurIPS.cc/2022/Conference — NeurIPS 2022 Accept_

### Official Review · Reviewer_9Pp5 · 2022-07-09

**Rating:** 6
**Confidence:** 4
**Soundness:** 3 good
**Presentation:** 2 fair
**Contribution:** 3 good

**Summary:**

This paper studies the generalization bounds of equivariant networks using the PAC-Bayes framework. The principal contribution of this work is an extension of the generalization bound from Neyshabur et. al 2018 to incorporate equivariant architectures by leveraging group and representation theory which allow the authors to characterize the bound using irreducible representations and their multiplicities. The authors also provide an empirical investigation of their generalization bounds on synthetic datasets as well variations of the MNIST dataset.

**Questions:**

From a technical point of view, the authors primarily work with $\rho_{\text{reg}}$ in the experiments which amount to the use of group convolutions as in Cohen et. al 2016. How does the current generalization theory change when we move to Steerable G-CNNs that $\textit{do not}$ use these representations but are built using say irreps, quotient reps, etc ... Secondly, another important symmetry group for ML purposes is $\mathcal{S}_n$ which is the symmetric group on $n$-elements. These can be used to define general permutation equivariant networks [1]. Can the authors bridge how their theory applies to these networks?


[1] H. Maron, H. Ben-Hamu, N. Shamir, and Y. Lipman. Invariant and equivariant graph networks. In
ICLR, 2019

**Strengths And Weaknesses:**

The main enjoys many strengths, the first of which is that the main contribution is presented in a clear and coherent way. Characterizing the generalization properties of equivariant networks is both an interesting direction and a needed one as the role of symmetry as powerful inductive biases are missing in the generalization literature. The main theorem and surrounding lemmas seem to be correct---although I did not check the technical details of Theorem 4.4 for correctness in the appendix as I did for the lemmas---and provide a sharp reduction in the hypothesis space and thus give tighter bounds.


The main weakness of this paper is that it seems that the authors missed the page limit for this year's NeurIPS is 9 pages and the current draft of the paper is considerably shorter (7.5 pages roughly). This is problematic because there are certain areas in the writing which could benefit from more detail. Chief among these is the experimental section, the plots in Fig 1 and 2 are incomprehensible and their explanations in the text don't add important details. For example, it seems row 2 in Fig 1 has a smaller generalization bound but the caption says that this is the bound obtained from a naive application by Neyshabur et. al 2018? There are also minor grievances that add to an overall lack of polish such as Equation 18 referenced in the main text to be an inequality is not the correct one being referenced in the appendix. There are also broken references (e.g. line 240 in the main text) and minor grammar mistakes all over. Given the additional page limit I encourage the authors to do the following, expand the experiment section and add more clarity. Add the details or bound of Appendix E to the main text as a way to contrast the tighter bound derived in the main text. Finally, provide a better proof sketch of the main theorem to the main text as much of the detail is completely committed (e.g. $\eta$ is not defined in the theorem). My current score is a reflection of the current status of this paper which appears to be a bit rushed and not complete. If the authors can adequately improve upon these the score may change accordingly.

---

> ### Author Response · Authors · 2022-08-02
> **On the presentation, and comparison with Neyshabur et. al 2018**
>
> We would like to thank the reviewers for the detailed comments. Please find below our answers.
>
> * Following other comments, we will update the structure of the paper according to the suggestions, particularly adding a sketch of proof, extended section on experiments with less cluttered figures, and starting with a simple and understandable version of equivariant networks and then relegate more technical details to the appendix. We will improve the presentation of the paper considering all these feedbacks.
>
> * In general, our bound is applicable to any equivariant networks with layers representable as decomposition of irreps of a compact group. This presupposes a vector space structure at each layer and compact group operation. With steerable networks, we need to carefully extend our analysis to case with noncompact groups. Our bound, however, should be applicable to graph neural networks given the compactness of cyclic group. Note that each operation in graph neural network needs to be represented in term of irreps.
>
> * To compare with Neyshabur et. al 2018, we used the bounds from section E, which ignores constant factors. On the other hand, these constant factors are considered for our bound. As we argued in section E, it can be analytically shown that our bound is tighter in general.

---

> > ### Comment · Reviewer_9Pp5 · 2022-08-09
> > **Re: Rebuttal**
> >
> > I thank the authors for their responses to my questions in the review. I was waiting to see if the authors would have time to incorporate my feedback and update the manuscript in time for a cleaner presentation. At the moment, I do not see any updates and as a result, I will maintain my current score despite noting that this paper has many interesting ideas which would benefit the community at large.

---

> > > ### Author Response · Authors · 2022-08-09
> > > **Response to the Reviewer 9Pp5**
> > >
> > > We have revised our paper by incorporating your feedback as well as other feedbacks, We made sure that Figures are understandable and the presentation has improved. Hopefully, the new version has addressed some of the concerns. Please find the new versions of the main paper and the supplementary materials both uploaded here. We are sorry for delay in uploading the revised version.

---

> > > > ### Comment · Reviewer_9Pp5 · 2022-08-09
> > > > **Re: Updated manuscript**
> > > >
> > > > Thank you for updating the manuscript. I will give this a fresh read and update my evaluation in the next few days accordingly.

---

### Official Review · Reviewer_MbmP · 2022-07-13

**Rating:** 6
**Confidence:** 2
**Soundness:** 3 good
**Presentation:** 3 good
**Contribution:** 3 good

**Summary:**

This work introduces a novel PAC-Bayes bound for equivariant networks in which the feature transformation is determined in terms of groups irreps.
The authors propose experiments in which the bound is shown to correlate with the generalization error, and the effect of group size, multiplicity and degree of the layers’ irreps is studied.

**Questions:**

I wonder if the assumption of homogeneity is too restricive. Can a bound be derived while relaxing this condition?

I would invite the authors to include some more relevant literature in their related work section. For instance:
“Generalization bounds for deep learning”, Valle-Perez and Louis and reference therein.

**Limitations:**

The authors point out that, experimentally, their bounds do not decrease with the size of the training size, a fact that should be actually expected. Do the authors have a theoretical explanations/hypothesis for it?

**Strengths And Weaknesses:**

The paper, despite the technical nature of the topic, is mostly clearly written. The assumptions are clearly stated and the proofs in the appendix relatively easy to follow, although I did not check every single step. The paper addresses an interesting topic, that is, the derivation of PAC-Bayes bound for equivariant networks.

I find it relatively hard to understand the motivation behind some technical assumptions (see section below for an example) about the setting for proving the theorem. I would appreciate if the authors could provide some more motivation behind their choices, or highlight choices that are made purely for a technical standpoint (i.e., it is possible to prove).

---

> ### Author Response · Authors · 2022-08-02
> **On homogeneity assumption and other comments**
>
> We would like to thank the reviewer for the encouraging comments. Please find below our answers.
>
> * The homogeneity can be relaxed to 1-Lipschitz and $\sigma(0)=0$. Actually, the generalization bound for this case is given in page 26, equation (50).
>
> * We will definitely update our references based on the suggestions, although it is difficult to do justice to sheer amount of works on this topic.
>
> * The bound not decreasing is scalable way with the training size is a general shortcoming of norm-based bounds as mentioned in Nagarajan and Kolter 2019. To summarize, they mention that existing norm based bounds have certain limitations in explaining generalization of neural networks. Our bound, being norm based, has a similar limitation. However, we mention that the bound can still provide useful guidelines for design and understanding of equivariant neural networks.

---

### Official Review · Reviewer_fd7r · 2022-07-15

**Rating:** 6
**Confidence:** 3
**Soundness:** 2 fair
**Presentation:** 2 fair
**Contribution:** 3 good

**Summary:**

The authors provide a generalization bound for a class of equivariant neural networks. The generalization bound extends prior bounds in the PAC Bayes literature. The prior PAC Bayes bounds were achieved by applying perturbations to weight matrices and studying the overlap of the posterior with the prior in the standard PAC Bayes setting. Here, perturbations are applied to parameters in the Fourier regime, which is parameterized much more sparsely than the real regime due to properties of linear equivariant layers as a consequence of Schur's Lemma. Their results indicate that the generalization bound they construct scales better with "group-related" parameters and may help explain generalization for invariant and equivariant models.

**Questions:**

- Have the authors compared to existing generalization bounds for CNNs? This could be a nice empirical analysis to compare to a special case of equivariant models.
- Are the y-axis values in figure 1 and figure 2 interpretable in any way. It seems that the complexity quantity is actually an order of magnitude or more smaller for the non-equivariant bound. Perhaps constant factors are not included here? This is concerning if the authors are trying to argue that the equivariant generalization bound outperforms the non-equivariant one it is inspired from.
- I assume results here only hold for finite groups? If so, this should be stated explicitly.
- Do bounds here inherently rely on invariance in the underlying model class one wants to learn? For example, if the final layer is fully connected making the network not invariant to the group action, does the bound suffer?
- From my understanding, it seems the bound is stronger when the dimension of an irrep in the parameterization is large and the multiplicities are small. Is this correct? On a related point, when all mulitplicities are $1$ and all irreps are dimension $1$ like in standard convolution over cyclic group, is there any advantage to the proposed bound in theorem 4? It seems that case reduces to the Neyshabur et al. bound as far as I can tell. If so, this would be nice to point out for intuition. Also, it would be interesting to explore whether architectures used in practice actually follow this type of structure.

**Limitations:**

All limitations are stated in other parts of this review.

**Strengths And Weaknesses:**

I was asked to review this paper under short notice and only over a couple of days; therefore, I was only able to read through the main paper. Proofs and mathematical details were not checked.

Overall, I commend the authors for taking on the task of obtaining generalization bounds for equivariant neural networks. A theoretical understanding of GCNNs is mostly lacking and the results here would take a relatively big step in helping to explain why generalization occurs. The results are definitely new and interesting, but I fear at times the organization and presentation of the paper lets it down. In the short time I had to review this paper, I struggled to understand the main theorem enough to really analyze it in full. The notation and basic setting also took too long for me to comprehend. The mathematical results in this paper are potentially very strong, but I think the authors need to re-organize and write the paper more cleanly for the insights to come through. For this reason, I slightly lean towards rejecting the paper, but welcome a discussion from the authors. Since I only have had a very short time to review this paper, I am open to changing my mind.

Strengths:
- Generalization bounds, specifically for GCNNs, are hard to find. I, at least, am not familiar with any that specifically look at the architectures the authors analyze. To me, this is an original contribution of the authors' paper.
- It can be challenging to analyze GCNNs theoretically since the group equivariance property adds yet another constraint to the theoretical analysis. The overall framework and mathematical tools the authors use to attack the problem may be applicable beyond the specific problem of generalization that the authors study.

Weaknesses:
- As a general point, there is little in the way of examples or intuition in this paper. The formula for the generalization bound is quite long. To add onto this, there are multiple norms within the generalization bound which can be challenging to understand even for readers who are familiar with GCNN literature. Comparing this to the bound in the Neyshabur et al. paper, it seems the major difference is in the Frobenius norm term with the dimension of the irrep dividing this factor. Is there anything else that has changed significantly? Is there a simple example where the authors can mathematically show their bound improves upon the Neyshabur et al. bound? For example, what happens with the cyclic group where we know all irreps are dimension 1?
- After reading through the choice of equivariant neural network in section 3.2 (and a brief foray into appendix B), it took me a very long time to understand the setup of the basic equivariant layer. As I understand, the authors chose the given parameterization to be as general as possible with their equivariant layer. However, it is very challenging to understand this parameterization without digging into the details of the appendix. It would be nice if this part of the paper were self-standing and a visual example would help greatly to understand notation.
- It seems that the equivariant parameters here are taken in Fourier space. This may be needed for proofs but as far as I can tell, this is only way to parameterize layers and not necessarily the most common. For example, even in standard convolution (i.e. over cyclic group), filters are parameterized over a sparse set of local elements. This is not necessarily "easy" to reconstruct in the Fourier domain as it would require enforcing parameter sharing across frequencies (i.e. just training over parameters in Fourier space will break the locality). Resulting perturbations to obtain generalization bounds may also ignore this locality in this case as enforcing perturbations in Fourier regime will not maintain the locality of the filter. The authors should at least comment on this point and be clear about the parameterization and its relevance to the models used in practice. This, after all, has obvious implications on generalization.


Smaller comments:
- The authors should at least state one commonly used architecture which follows the parameterization that they have selected both to aid intuition and to better ground the findings.
- Figure 3 is hard to read. The font size is very small and the 3-D plots are hard to follow.
- Similarly, figure 2 is quite crowded and in need of simplification and larger font.
- Line 188: [18] should be [7]?
- Line 240: section reference ??
- Line 218: two periods
- $\beta$ in equation 12 is raised to the power $2L$ but then in the definition of $\beta$, it is taken to the power $1/L$. Why not just ignore the added $L$ factor?




My recommendations to the authors on how to make the paper more readable:
- Adjust section 3.2 to make it clearer how equivariant operations are parameterized. Start from basic definitions of convolution (e.g. in Fourier or real regimes) and then explain how the parameterization that they choose is equivalent to a specific parameterization in the Fourier or real regime. Give an example of an existing architecture to aid intuition. State whether parameterization in the form analyzed can be restricted locally (e.g., restricting to sparse set of irreps is not necessarily equivalent to a sparse real support as stated earlier). Some of the information currently there can be relegated to appendix (e.g. only look at $c_\psi=1$ case and relegate rest to appendix).
- Provide intuition behind the generalization bound by perhaps bounding further parameters or looking at a special case where some of the terms become constant. E.g. fix a certain number of layers and then look at the dependence on the other parameters.
- Figures are too crowded right now. it is hard to read any of the figures as they stand and they take up too much space. I would have preferred to have more details on the architecture used to train networks and how that architecture varied for the various points in the figures. Also, a more complete description of how the comparison bound was calculated would be helpful.



---
**Post-feedback changes**
I am upgrading my score to a weak accept after the authors' valuable update to the paper and their comments to my feedback. I still have some concerns about the presentation of the paper and the limitations which I have listed in my response to the authors' comments below.

---

> ### Author Response · Authors · 2022-08-02
> **Answer to the questions of the reviewer**
>
> We would like to thank the reviewer for the detailed comments. Please find below our answers.
>
> **Questions:**
>
> * We have not compared with bounds on CNNs. Vanilla CNNs are translation equivariant, which is a non-compact group. The current results need to be extended and adjusted to those cases.
> * The dependency on degree, multiplicity and dimension of irreps is a bit trickier and captured by $M(l,\eta)$ (see Discussion on page 4). The main reason behind the difference in numerical results with Neyshabur et al, as discussed also below, is the presence of constant factors. We had a section in our Supplementary material regarding Neyshabur et al. paper (Section E). The main difference comes from  conducting our analysis in the Fourier domain, which has impact on the tightness of the bound. It was shown applying the Neyshabur et al. bound directly would not quantify properly the impact of group size on generalization. This holds even for the cyclic group, and generally Abelian groups, with  irreps of dimension 1.
>
> * The result holds for compact groups, as we mentioned in page 3. We make it more explicit in the revision.
>
>
> * The bound is currently about equivariant networks. However, the proof technique applies to hybrid architectures with FCN appended at the end. The generalization error would be different in those cases.
>
> * Regarding the presentation of the paper, it is admittedly challenging to present in a unified and accessible way tools from geometric deep learning, namely representation and group theory, and tools from statistical learning theory. Given the comments from other reviewers, we try to improve the presentation concretely by having a simplified version of the theorem in the paper with milder notations,  presenting equivariant networks in a step by step way relying on familiar convolutional setups and supporting it with visualization, and explicating further the intuition behind each theoretical term including the norms. Hopefully, we can make sure that our theoretical contribution is properly appreciated and understood.

---

> > ### Comment · Reviewer_fd7r · 2022-08-05
> > **Response and further questions**
> >
> > Thank you for the follow-up. The proposed changes to the paper would be very nice and if they can be made in time for me to check them, that would very much help sway me towards accepting the paper. Since I had to review the paper on short notice, I am taking another look now and am willing to reconsider my rating, but I still think some of my questions remain unanswered.
> >
> > Questions remaining:
> > - Can the authors confirm that the parameterization here places all parameters on irreps as per the multiplicities given by Schur's Lemma? My comment about CNNs was for the finite cyclic group which corresponds to CNNs with cycling. This is a finite group and thus also compact. For example, when you parameterize locally in a standard CNN (as is pretty much always done), the parameters in Fourier space will have some sharing. As far as I can tell, your parameterization may "over-count" the number of parameters in such a setting as the setting you consider assumes full parameterization in Fourier space and essentially ignores this sharing. Is this correct or am I missing something? I don't think this is necessarily a serious issue, but it very much needs to be clarified since I think this can impact the tightness and relevance of the bound.
> > - This question still remains unanswered and I think this could be crucial to improve the presentation of the paper: Is there a simple example where the authors can mathematically show their bound improves upon the Neyshabur et al. bound? For example, it seems that the most significant term that changes from the Neyshabur et al. bound is the term corresponding to the ratio of Frobenius to spectral norms. Is this correct and if so, can you make a comparison between your bound and the existing bound for a simple architecture?
> >
> >
> > As a smaller comment, on my second read, there is another point of smaller concern that I want to ask about. Are you restricting irreps to be real only? If so, this should be clarified as it can impact dimensions of representations and the decomposition of representations into irreps. Furthermore, there is one small typo in line 221 where you say "all real irreps are 2-dimensional". Technically, this should say all real irreps are at most 2-dimensional since there definitely exist one dimensional real irreps (e.g., the trivial one).

---

> > > ### Author Response · Authors · 2022-08-09
> > > **Response to the reviewer fd7r**
> > >
> > > We would like to thank the reviewer for the comments and questions. Please find our answers below.
> > >
> > > * Thanks for clarifying the question. We confirm that our parametrization utilizes decomposition of each feature space as a direct sum of irreps and then Schur’s lemma for defining the weights. For the case of CNNs, the weight sharing, namely the convolution, leads to a concise point-wise multiplication in Fourier space. Mathematically, the number of parameters in ambient and Fourier spaces remain the same. However, narrower receptive field of convolution kernels in vanilla CNN is an additional inductive bias that further reduces the number of parameters. We have not considered this in our work. I hope we did not misunderstand the question.
> > > * We added a whole new section clarifying situations in which the bound is tighter. Please see page 7 and 8 of the new uploaded paper. To summarize, there are two main differences w.r.t Neyshabur et al., first is the ratio of Frobenius to spectral norms, and the second is the dimension dependency related to the term $M(l,\eta)$. The norm ratio for our bound is strictly less than Neyshabur et al. when $\dim_\psi > 1$. We provided examples for which the other terms are also tighter than Neyshabur et al.
> > > * Since deep learning implementations are all done predominantly in real domain, we focus on real irreps. As you mentioned, this impacts their dimension and leads to the parameter $c_\psi$ and categorization of real irreps, as we had explained in  the supplementary materials. The theory aims at characterization of generalization error in terms of the type of irreps used.
> > > * Thanks for pointing the typo regarding 2-dimensional irreps. We have fixed it in the new version.
> > >
> > > Finally, we would like to emphasize that we have uploaded a new version (the main paper and supplementary materials) addressing current requests.

---

> > > > ### Comment · Reviewer_fd7r · 2022-08-10
> > > > **Final comments**
> > > >
> > > > I thank the authors for updating their paper. This version is indeed much better. Some comments/concerns still remain which can be addressed in the final version:
> > > > - Requiring parameters in Fourier space needs to be clearly listed as an assumption and potentially even a limitation. You state this in your paper, e.g. saying “by letting the transformations acting only on some of the frequencies”. This however is an assumption and the language here makes it seem like it is not. When parameterization is performed in real space, the bound here may not apply or can be very poor.
> > > > - The visualization in figure 1 needs improving - the linear vector space structure is difficult to observe there. Using block diagonal matrices would help rather than the choice of structure shown here. I understand the authors had limited time, but I think this would be great to improve when given time for the final paper.
> > > > - Many networks do actually use complex parameters which changes the bound, e.g. see Spherical CNN. This needs to be addressed or at least stated as an assumption/limitation clearly.
> > > >
> > > > Some smaller comments about presentation:
> > > > - Figures still a bit cluttered in the experiments section. I don’t think there needs to be as many colors for example.
> > > > - I personally do not like the 3D plots since the angles can make it hard to read the exact patterns. I would just put separate 2D plots showing the correlations though this is just my opinion.
> > > > - line 135: unfinished sentence
> > > > - line 213: typo “the” line 280: unneeded comma “as long as,”
> > > > - line 305: “can not”
> > > > - throughout: supplementary materials is sometimes capitalized and sometimes not

---

### Official Review · Reviewer_rgnq · 2022-07-18

**Rating:** 4
**Confidence:** 4
**Soundness:** 2 fair
**Presentation:** 2 fair
**Contribution:** 2 fair

**Summary:**

This paper gives generalization bounds on equivariant models for general groups.
Generalization bounds for equivariant models in the case of finite groups were given by Sannai et al. but generalization bounds for general groups were not known. The boundary is described by an invariant called multiplicity.
This attempt is new to the best of my knowledge.


**Questions:**

According to Sannai et al. the generalization gap is obtained by the number of coverings of the quotient space for the group action. Can you see that it is tighter than its bounds?

**Limitations:**

Properly addressed.

**Strengths And Weaknesses:**

The author should first note and cite that generalization bounds are given for finite groups by Sannai et al.

Strengths
This paper differs from Sannai et al. in that it deals with equivariant models that are not limited to finite groups.
Furthermore, the bounds in this paper use information on the multiplicity of irreducible representations used in the model. This is commendable.

Weaknesses
On the other hand, the right-hand side still has some parts that have not been fully calculated, such as multiplicities.
As it is, I do not believe that any useful findings have been obtained.
Overall, the results are not considered significant.

---

> ### Author Response · Authors · 2022-08-02
> **On different focuses of our work and Sannai et al. and improvements with respect to their paper**
>
> We would like to thank the reviewer for the comments. Please find below our answers.
> * The degree and multiplicities of irreps are design choices. This means that they are fully determined for each network realization. One can roughly compare them to the size and number of convolutional filters. One the main contributions of our work is to characterize the impact of these parameters on the generalization error. Therefore, our bound is fully computable for equivariant networks built using irreps, as it depends on different norms and the network architecture.
>  * Our bound can provide guidelines about how to choose degrees and multiplicities of irreps for better generalization. This is a new finding and can be useful for practitioners. We discuss this point in page 6, before Experiments section.
>  * We had mentioned and cited Sannai et al in our submitted paper. In light of reviewers comments, we will extend the discussions to include more details about it. We would like to clarify that the focus of their paper is on the impact of general invariance and equivariance on the generalization error, where they report the improvement in scaling. In particular, they mention in Remark, 1 that they are occupied with characterizing invariance and do not consider specific bounds on the covering number. In contrast, we are interested in obtained architecture dependent generalization bounds with the same group invarariance.
>  * It is difficult to compare the  tightness of our bound with respect to Sannai et al., since, as mentioned above, they focus on the scaling. On that note, however, Lemma 2 of Sannai et al. shows the scaling of generalization error with group size. We report a similar scaling in our paper and discuss it numerically and theoretically (Fig.3, Section F.2, discussions in page 27-29). As an improvement, our generalization bound depends on the sample size as $O(1/\sqrt{m}$, which is better than $O(1/\sqrt{m^{2/n}})$ of Sannai et al.

---

### Author Response · Authors · 2022-08-09
**A new version of the paper is uploaded**

Upon the request of the reviewers, we have uploaded a new version including most of the requested changes. Of course, we moved some discussions to supplementary materials to remain within 9 pages limit, however, we can still move them to the main body of the paper for the camera ready and make it within its 10 pages limit.
Among others, we had changed the following items:
* We have added more comments on Sanai et al.
* We have a simplified version of the theorem in the paper with milder notations.
* We have presented equivariant networks starting with an example and using new visualization (Fig 4 and Fig 5).
* We have added more comments on the intuition behind each theoretical term including the norms.
* We have added more detailed discussions on the comparison with Neyshabur et al.
* We have added new experiments for connecting $M(l,\eta)$ to the generalization error (see Section F.6 and Figure 15).
* We have updated our references based on the reviewers’ suggestions.
* We have added a short sketch of proof with its steps.
* We have modified the experiment section with clearer figures.

---

### Meta-Review · Area_Chair_SMht · 2022-08-27

**Recommendation:** Accept
**Confidence:** Less certain

**Metareview:**

This is a borderline paper studying an interesting question around generalization bounds for equivariant networks. Initially there were significant concerns around presentation of the key results and related work. During the rebuttal phase authors updated the manuscript as per reviewers suggestions, resulting in a significantly better manuscript. I applaud the efforts of all the reviewers who engaged with authors leading to a better submission. One reviewer still kept their negative score, but other reviewers and I believe their concerns were addressed in the updated manuscript.

Overall I recommend acceptance and it is important that the authors revise the manuscript highlighting the key assumptions upfront about the Fourier space following Reviewer fd7r's suggestions.

**Award:**

No

---

### Decision · Program_Chairs · 2022-09-14

Accept